# Debiased Collaborative Filtering with Kernel-Based Causal Balancing

**Haoxuan Li**[1]    **Chunyuan Zheng**[1]    **Yanghao Xiao**[2]    **Peng Wu**[3]    **Zhi Geng**[3]
**Xu Chen**[4,*]    **Peng Cui**[5]
[1]Peking University    [2]University of Chinese Academy of Sciences
[3]Beijing Technology and Business University    [4]Renmin University of China
[5]Tsinghua University
hxli@stu.pku.edu.cn   xu.chen@ruc.edu.cn   cuip@tsinghua.edu.cn

## ABSTRACT

Debiased collaborative filtering aims to learn an unbiased prediction model by removing different biases in observational datasets. To solve this problem, one of the simple and effective methods is based on the propensity score, which adjusts the observational sample distribution to the target one by reweighting observed instances. Ideally, propensity scores should be learned with causal balancing constraints. However, existing methods usually ignore such constraints or implement them with unreasonable approximations, which may affect the accuracy of the learned propensity scores. To bridge this gap, in this paper, we first analyze the gaps between the causal balancing requirements and existing methods such as learning the propensity with cross-entropy loss or manually selecting functions to balance. Inspired by these gaps, we propose to approximate the balancing functions in reproducing kernel Hilbert space and demonstrate that, based on the universal property and representer theorem of kernel functions, the causal balancing constraints can be better satisfied. Meanwhile, we propose an algorithm that adaptively balances the kernel function and theoretically analyze the generalization error bound of our methods. We conduct extensive experiments to demonstrate the effectiveness of our methods, and to promote this research direction, we have released our project at https://github.com/haoxuanli-pku/ICLR24-Kernel-Balancing.

## 1 INTRODUCTION

Collaborative filtering (CF) is the basis for a large number of real-world applications, such as recommender system, social network, and drug repositioning. However, the collected data may contain different types of biases, which poses challenges to effectively learning CF models that can well represent the target sample populations (Marlin and Zemel, 2009). To solve this problem, people have proposed many debiased CF methods, among which propensity-based methods are simple and effective, which adjust the observational sample distribution to the target one by reweighting observed instances. For example, Schnabel et al. (2016) proposes to use the inverse propensity score (IPS) to reweight the observed user-item interactions. The doubly robust (DR) method is another powerful and widely-used propensity-based method for debiasing, which combines an imputation model to reduce the variance and achieve double robustness property, *i.e.*, unbiased either the learned propensity scores or the imputed errors are accurate (Wang et al., 2019).

Despite previous propensity-based methods have achieved many promising results, most of them ignore the causal balancing constraints (Imai and Ratkovic, 2014; Li et al., 2018; 2023d), which has been demonstrated to be important and necessary for learning accurate propensities. Specifically, causal balancing requires that the propensity score can effectively pull in the distance between the observed and unobserved sample for *any* given function $\phi(\cdot)$ (Imai and Ratkovic, 2014), that is

$$\mathbb{E}\left[\frac{o_{u,i}\phi(x_{u,i})}{p_{u,i}}\right] = \mathbb{E}\left[\frac{(1-o_{u,i})\phi(x_{u,i})}{1-p_{u,i}}\right] = \mathbb{E}[\phi(x_{u,i})],$$

---

*Corresponding author.

where $x_{u,i}$ is the covariate of user $u$ and item $i$, $o_{u,i}$ indicates whether the outcome of user $u$ to item $i$ is observed, and $p_{u,i} := \mathbb{P}(o_{u,i} = 1|x_{u,i})$ is the propensity score.

Existing debiased CF methods usually learn the propensity score based on two strategies: (1) adopting cross-entropy to train $p_{u,i}$ that predicts $o_{u,i}$ using all user-item pairs (Wang et al., 2019), which does not consider the balancing property; (2) using the above causal balancing constraint to learn $p_{u,i}$ with finite manually selected balancing functions $\phi(\cdot)$ (Li et al., 2023e). However, the selected balancing functions may not be a good proxy of all functions, leading to insufficient balancing.

To bridge the above gaps, we propose a debiased CF method that can adaptively capture functions that are more in need of being balanced. Specifically, we first analyze the relations between the causal balancing constraints and previous propensity score learning methods, motivating our research from a novel perspective. Then, to achieve the balancing property for any $\phi(\cdot)$, we propose to conduct causal balancing in the reproducing kernel Hilbert space (RKHS), where any continuous function can be approximated based on Gaussian or exponential kernels. Moreover, we design a kernel balancing algorithm to adaptively balance the selected functions and theoretically analyze the generalization error bounds. Note that the proposed kernel balancing method applies to both pure propensity-based and DR-based methods. The main contributions of this paper can be concluded as follows

• We theoretically prove the unbiasedness condition of the propensity-based methods from the function balancing perspective, revealing the shortcomings of previous propensity learning methods using cross-entropy and manually specified balancing functions.

• We design a novel kernel balancing method that adaptively find the balancing functions that contribute the most to reducing the estimation bias via convex optimization, named adaptive kernel balancing, and derive the corresponding generalization error bounds.

• We conduct extensive experiments on three publicly available datasets to demonstrate the effectiveness of the proposed adaptive kernel balancing approach for IPS and DR estimators.

## 2 PRELIMINARIES

### 2.1 DEBIASED COLLABORATIVE FILTERING

Let $\mathcal{U}$ and $\mathcal{I}$ be the whole user and item sets, respectively. Denote $\mathcal{U}_o = \{u_1, \ldots, u_m\} \subseteq \mathcal{U}$ and $\mathcal{I}_o = \{i_1, \ldots, i_n\} \subseteq \mathcal{I}$ be the observed user and item sets randomly sampled from the super-population, and $\mathcal{D} = \mathcal{U}_o \times \mathcal{I}_o = \{(u,i) \mid u \in \mathcal{U}_o, i \in \mathcal{I}_o\}$ be the corresponding user-item set. For each user-item pair $(u,i) \in \mathcal{D}$, we denote $x_{u,i} \in \mathbb{R}^K$, $r_{u,i} \in \mathbb{R}$ and $o_{u,i} \in \{0,1\}$ as the user-item features, the rating of user $u$ to item $i$ and whether $r_{u,i}$ is observed in the dataset, respectively. For brevity, denote $\mathcal{O} = \{(u,i) \mid (u,i) \in \mathcal{D}, o_{u,i} = 1\}$ as the set of user-item pairs with obseverd $r_{u,i}$.

Let $\hat{r}_{u,i} = f(x_{u,i}; \theta_r)$ be the prediction model parameterized by $\theta_r$, which predicts $r_{u,i}$ according to the features $x_{u,i}$. To achieve the unbiased learning, it should be trained by minimizing the ideal loss:

$$\mathcal{L}_{\text{Ideal}}(\theta_r) = \frac{1}{|\mathcal{D}|} \sum_{(u,i) \in \mathcal{D}} e_{u,i},$$

where $e_{u,i} = \mathcal{L}(\hat{r}_{u,i}, r_{u,i})$ is the prediction error with $\mathcal{L}(\cdot, \cdot)$ be an arbitrary loss function, e.g., mean square loss or cross-entropy loss. However, observing all $r_{u,i}$ is impractical so that $e_{u,i}$ is not computable for $o_{u,i} = 0$. A naive method for solving this problem is approximating $\mathcal{L}_{\text{Ideal}}(\theta_r)$ directly based on the observed samples, that is to minimize the naive loss

$$\mathcal{L}_{\text{Naive}}(\theta_r) = \frac{1}{|\mathcal{O}|} \sum_{(u,i) \in \mathcal{O}} e_{u,i}.$$

However, due to the existence of selection bias, $\mathcal{L}_{\text{Naive}}(\theta_r)$ is not unbiased in terms of estimating $\mathcal{L}_{\text{Ideal}}(\theta_r)$ (Wang et al., 2019). To further build unbiased estimators, previous studies propose to use propensity score to adjust observed sample weights, and design the IPS loss

$$\mathcal{L}_{\text{IPS}}(\theta_r) = \frac{1}{|\mathcal{D}|} \sum_{(u,i) \in \mathcal{D}} \frac{o_{u,i} e_{u,i}}{\hat{p}_{u,i}},$$

where $\hat{p}_{u,i} = \pi(x_{u,i}; \theta_p)$ is the estimation of propensity score $p_{u,i}$. It can be demonstrated that $\mathcal{L}_{\text{IPS}}(\theta_r)$ is unbiased when $p_{u,i} = \hat{p}_{u,i}$ (Schnabel et al., 2016; Wang et al., 2019). To further improve the robustness and reduce the variance, researchers extend the IPS method to many DR methods (Wang et al., 2019; 2021; Li et al., 2023b;f) with the DR loss

$$\mathcal{L}_{\text{DR}}(\theta_r) = \frac{1}{|\mathcal{D}|} \sum_{(u,i) \in \mathcal{D}} \left[ \hat{e}_{u,i} + \frac{o_{u,i} \cdot (e_{u,i} - \hat{e}_{u,i})}{p_{u,i}} \right],$$

where $\hat{e}_{u,i} = m(x_{u,i}; \theta_e)$ is the imputed error. The DR estimator is unbiased when all the estimated propensity scores or the imputed errors are accurate. In both the IPS and DR methods, computing the propensity score is of great importance, which directly determines the final debiasing performance.

## 2.2 CAUSAL BALANCING

In many causal inference studies (Imai and Ratkovic, 2014; Imbens and Rubin, 2015; Rosenbaum, 2020; Sant'Anna et al., 2022), accurately computing the propensity score is quite challenging, since it is hard to specify the propensity model structure and estimate the model parameters. To solve this problem, researchers propose a general strategy to learn the propensity scores without specifying model structure based on the following causal balancing constraints (Imbens and Rubin, 2015)

$$\mathbb{E}\left[ \frac{o_{u,i} \phi(x_{u,i})}{p_{u,i}} \right] = \mathbb{E}\left[ \frac{(1 - o_{u,i}) \phi(x_{u,i})}{1 - p_{u,i}} \right] = \mathbb{E}[\phi(x_{u,i})], \tag{1}$$

where $\phi : \mathcal{X} \rightarrow \mathbb{R}$ is a balancing function applied to the covariant. Ideally, this equation should hold for *any* balancing function. Inspired by such property, a recent work (Li et al., 2023e) proposes to learn the propensity by minimizing the distance between the first and second terms in Equation (1). However, in this method, the finite balancing functions $\phi(\cdot)$ are manually selected (*e.g.*, the first and second moments), which may not be a good proxy of all functions, leading to insufficient balancing.

## 3 CONNECTING CAUSAL BALANCING AND EXISTING PROPENSITY LEARNING

In the field of debiased collaborative filtering, there are usually two types of propensity score learning methods: (1) using cross-entropy to train $p_{u,i}$ that predicts $o_{u,i}$ using all user-item pairs; (2) adopting the causal balancing method with a finite number of manually selected balancing functions $\phi(\cdot)$.

### 3.1 CROSS-ENTROPY BASED STRATEGY

Recall that the propensity model $\pi(x_{u,i}; \theta_p)$ aims to predict the probability of observing $r_{u,i}$ in the dataset (*i.e.*, $o_{u,i} = 1$). The cross-entropy based strategy learns $\theta_p$ based on the following loss

$$\mathcal{L}_p(\theta_p) = \sum_{(u,i) \in \mathcal{D}} -o_{u,i} \log \{ \pi(x_{u,i}; \theta_p) \} - (1 - o_{u,i}) \log \{ 1 - \pi(x_{u,i}; \theta_p) \}.$$

By taking the first derivative of this loss function *w.r.t* $\theta_p$, the optimal $\pi(x_{u,i}; \theta_p)$ should satisfy

$$\frac{\partial \mathcal{L}_p(\theta_p)}{\partial \theta_p} = \sum_{(u,i) \in \mathcal{D}} -\frac{o_{u,i} \partial \pi(x_{u,i}; \theta_p)/\partial \theta_p}{\pi(x_{u,i}; \theta_p)} + \frac{(1 - o_{u,i}) \partial \pi(x_{u,i}; \theta_p)/\partial \theta_p}{1 - \pi(x_{u,i}; \theta_p)} = 0. \tag{2}$$

By comparing this requirement with the causal balancing constraint in Equation (1), we can see that if we let $\phi(x_{u,i}) = \partial \pi(x_{u,i}; \theta_p)/\partial \theta_p$, then Equation (2) is a special case of Equation (1), which means that the cross-entropy based strategy is not sufficient to achieve causal balancing.

### 3.2 CAUSAL BALANCING WITH MANUALLY SPECIFIED BALANCING FUNCTIONS

Li et al. (2023e) is a recent work on using causal balancing for debiased collaborative filtering. In this work, the authors first manually select and fix $J$ balancing functions $\{h^{(1)}(\cdot), h^{(2)}(\cdot), \ldots, h^{(J)}(\cdot)\}$. Denote $\hat{w}_{u,i} = g(x_{u,i}; \theta_w)$ be the balancing weight assigned to sample $(u, i)$, then the objective

function and constrains of the optimization problem for learning $\theta_w$ is shown below

$$\max_{\theta_w} - \sum_{(u,i) \in \mathcal{O}} \hat{w}_{u,i} \log \hat{w}_{u,i}$$

$$\text{s.t. } \frac{1}{|\mathcal{D}|} \sum_{(u,i) \in \mathcal{D}} o_{u,i} \hat{w}_{u,i} h^{(j)}(x_{u,i}) = \frac{1}{|\mathcal{D}|} \sum_{(u,i) \in \mathcal{D}} h^{(j)}(x_{u,i}) \qquad j \in \{1, \ldots, J\},$$

$$\frac{1}{|\mathcal{D}|} \sum_{(u,i) \in \mathcal{D}} o_{u,i} \hat{w}_{u,i} = 1, \qquad \hat{w}_{u,i} \geq 0 \qquad \forall (u,i) \in \mathcal{O},$$

where the objective aims to effectively avoid extremely small balancing weight via maximizing the entropy (Guiasu and Shenitzer, 1985). The first constraint is the empirical implementation of Equation (1) based on balancing functions $\{h^{(1)}(\cdot), h^{(2)}(\cdot), \ldots, h^{(J)}(\cdot)\}$ and the second constraint imposes normalization regularization on $\hat{w}_{u,i}$. Remarkably, this objective is convex *w.r.t.* $\hat{w}_{u,i}$, which can be solved by the Lagrange multiplier method. The following Theorem 1 shows the estimation bias depends on the distance between $e_{u,i}$ and $\mathcal{H}_J = \text{span}\{h^{(1)}(x_{u,i}), \ldots, h^{(J)}(x_{u,i})\}$.

**Theorem 1.** *If $e_{u,i} \in \mathcal{H}_J = \text{span}\{h^{(1)}(\cdot), \ldots, h^{(J)}(\cdot)\}$, then the above learned propensities lead to an unbiased ideal loss estimation in term of the IPS method.*

The balancing functions $\{h^{(1)}(\cdot), \ldots, h^{(J)}(\cdot)\}$ are manually selected in Li et al. (2023e), which is equivalent to letting $\phi(x_{u,i}) = h^{(j)}(x_{u,i})$, $j \in \{1, \ldots, J\}$ in Equation (1). This method improves the cross-entropy based strategy by using more balancing functions. However, the selected balancing functions may not well represent $e_{u,i}$, that is, $e_{u,i} \notin \mathcal{H}_J = \text{span}\{h^{(1)}(\cdot), \ldots, h^{(J)}(\cdot)\}$, which may lead to inaccurate balancing weights estimation and biased prediction model learning.

## 4 KERNEL-BASED CAUSAL BALANCING

### 4.1 KERNEL FUNCTION, UNIVERSAL PROPERTY, AND REPRESENTER THEOREM

To satisfy the causal balancing constraint in Equation (1), we approximate the balancing function with Gaussian and exponential kernels in the reproducing kernel Hilbert space (RKHS). To begin with, we first introduce several basic definitions and properties of the kernel function.

**Definition 1** (Kernel function). *Let $\mathcal{X}$ be a non-empty set. A function $K : \mathcal{X} \times \mathcal{X} \to \mathbb{R}$ is a kernel function if there exists a Hilbert space $\mathcal{H}$ and a feature map $\psi : \mathcal{X} \to \mathcal{H}$ such that $\forall x, x' \in \mathcal{X}$, $K(x, x') := \langle \psi(x), \psi(x') \rangle_{\mathcal{H}}$.*

Gaussian and exponential kernels are two typical kernel functions, which are formulated as follows

$$K^{\text{Gau}}(x, x') = \exp\left(-\frac{\|x - x'\|^2}{2\sigma^2}\right) \quad \text{and} \quad K^{\text{Exp}}(x, x') = \exp\left(-\frac{\|x - x'\|}{2\sigma^2}\right).$$

**Definition 2** (Universal kernel). *For $\mathcal{X}$ compact Hausdorff, a kernel is universal if for any continuous function $e : \mathcal{X} \to \mathbb{R}$ and $\epsilon > 0$, there exists $f \in \mathcal{H}$ in the corresponding RKHS such that $\sup_{x \in \mathcal{X}} |f(x) - e(x)| \leq \epsilon$.*

**Lemma 1** (Sriperumbudur et al. (2011)). *Both the Gaussian and exponential kernels are universal.*

This lemma shows that there is a function in RKHS $\mathcal{H} = \text{span}\{K(\cdot, x) \mid x \in \mathcal{X}\}$ that can approach any continuous function when the kernel function $K(\cdot, x)$ is chosen as the Gaussian or exponential kernel. However, $\mathcal{H}$ might be an infinity dimension space with $|\mathcal{X}| = \infty$, which leads to infinity constraints for the optimization problem. The following representer theorem guarantees the optimality of kernel methods under penalized empirical risk minimization and provides a form of the best possible choice of kernel balancing under finite samples.

**Lemma 2** (Representer theorem). *If $\Omega = h(\|f\|)$ for some increasing function $h : \mathbb{R}_+ \to \overline{\mathbb{R}}$, then some empirical risk minimizer must admit the form $f(\cdot) = \sum_{i=1}^{n} \alpha_i K(\cdot, x_i)$ for some $\boldsymbol{\alpha} = (\alpha_1, \ldots, \alpha_n) \in \mathbb{R}^n$. If $h$ is strictly increasing, all minimizers admit this form.*

## 4.2 WORST-CASE KERNEL BALANCING

Next, we propose kernel balancing IPS (KBIPS) and kernel balancing DR (KBDR) for debiased CF

$$\mathcal{L}_{\text{KBIPS}}(\theta_r) = \frac{1}{|\mathcal{D}|} \sum_{(u,i)\in\mathcal{D}} o_{u,i}\hat{w}_{u,i}e_{u,i},$$

$$\mathcal{L}_{\text{KBDR}}(\theta_r) = \frac{1}{|\mathcal{D}|} \sum_{(u,i)\in\mathcal{D}} \left[ \hat{e}_{u,i} + o_{u,i}\hat{w}_{u,i}(e_{u,i} - \hat{e}_{u,i}) \right], \tag{3}$$

where the balancing weights $\hat{w}_{u,i}$ are learned via either the proposed worst-case kernel balancing in the rest of this section or the proposed adaptive kernel balancing method in Section 4.3.

For illustration purposes, we use KBIPS as an example, and KBDR can be derived in a similar way[1]. Theorem 1 shows that when the prediction error function $e_{u,i} \in \mathcal{H}_J = \text{span}\{h^{(1)}(\cdot), \dots, h^{(J)}(\cdot)\}$ and the learned balancing weights can balance those functions $\{h^{(1)}(\cdot), \dots, h^{(J)}(\cdot)\}$, then the above KBIPS estimator leads to the unbiased ideal loss estimation. However, in practice, the prediction error function $e_{u,i}$ could be any continuous function, lying in a much larger hypothesis space than $\mathcal{H}_J$. By Lemma 1, when the kernel function $K(\cdot, x)$ is chosen as the Gaussian or exponential kernel, we can assume $e_{u,i} \in \mathcal{H} = \text{span}\{K(\cdot, x_{u,i}) \mid (u,i) \in \mathcal{U} \times \mathcal{I}\}$ holds with any approximation error $\epsilon$.

Note that the empirical bias of the KBIPS estimator for estimating the ideal loss is

$$\text{Bias}(\mathcal{L}_{\text{KBIPS}}(\theta_r)) = \{\mathcal{L}_{\text{KBIPS}}(\theta_r) - \mathcal{L}_{\text{Ideal}}(\theta_r)\}^2 = \left\{ \frac{1}{|\mathcal{D}|} \sum_{(u,i)\in\mathcal{D}} (o_{u,i}\hat{w}_{u,i} - 1)e_{u,i} \right\}^2,$$

then the worst-case kernel balancing (WKB) method focuses on controlling the worst-case bias of KBIPS by playing the following minimax game

$$\min_{\theta_w} \left[ \sup_{e\in\tilde{\mathcal{H}}} \left\{ \frac{1}{|\mathcal{D}|} \sum_{(u,i)\in\mathcal{D}} (o_{u,i}\hat{w}_{u,i} - 1)e_{u,i} \right\}^2 \right] = \min_{\theta_w} \left[ \sup_{e\in\mathcal{H}} \frac{\left\{ \frac{1}{|\mathcal{D}|} \sum_{(u,i)\in\mathcal{D}} (o_{u,i}\hat{w}_{u,i} - 1)e_{u,i} \right\}^2}{\frac{1}{|\mathcal{D}|} \sum_{(u,i)\in\mathcal{D}} e_{u,i}^2} \right],$$

where $\tilde{\mathcal{H}} = \{e(\cdot) \in \mathcal{H} : \|e(\cdot)\|_N^2 = |\mathcal{D}|^{-1} \sum_{(u,i)\in\mathcal{D}} e_{u,i}^2 = 1\}$ is the normalized RKHS. By the representer theorem in Lemma 2, the right-hand side is the same as the following

$$\min_{\theta_w} \left[ \sup_{\alpha_{s,t}} \frac{\left\{ \frac{1}{|\mathcal{D}|} \sum_{(u,i)\in\mathcal{D}} (o_{u,i}\hat{w}_{u,i} - 1) \sum_{(s,t)\in\mathcal{D}} \alpha_{s,t} K(x_{u,i}, x_{s,t}) \right\}^2}{\frac{1}{|\mathcal{D}|} \sum_{(u,i)\in\mathcal{D}} e_{u,i}^2} \right].$$

## 4.3 ADAPTIVE KERNEL BALANCING

There are $|\mathcal{D}|$ kernel functions in the above objective. Since there are usually a large number of users and items in the recommender systems, $|\mathcal{D}|$ is quite large, which makes it infeasible to balance all kernel functions. To solve this problem, a straightforward method is to randomly select $J$ functions from $\text{span}\{K(\cdot, x_{u,i}) \mid (u,i) \in \mathcal{D}\}$ to balance, named random kernel balancing (RKB). However, this method regards all kernel functions as equally important, which harms the debiasing performance.

To overcome the shortcomings of the WKB and RKB methods, we propose a novel adaptive kernel balancing (AKB) method that can adaptively select which kernel functions to balance. Given current prediction model $f(x_{u,i}; \theta_r)$, we first fit $e_{u,i}$ using the kernel functions in RKHS

$$(\alpha_{1,1}, \dots, \alpha_{m,n}) = \arg\min_{\boldsymbol{\alpha}} \frac{1}{|\mathcal{D}|} \sum_{(u,i)\in\mathcal{D}} \left\{ e_{u,i} - \sum_{(s,t)\in\mathcal{D}} \alpha_{s,t} K(x_{u,i}, x_{s,t}) \right\}^2, \tag{4}$$

---

[1] For KBDR, it requires that $e_{u,i} - \hat{e}_{u,i} \in \mathcal{H}_J = \text{span}\{h^{(1)}(\cdot), \dots, h^{(J)}(\cdot)\}$. Same as the follows, *e.g.*, $(\alpha_{1,1}, \dots, \alpha_{m,n})$ should minimize the mean squared error between $e_{u,i} - \hat{e}_{u,i}$ and $\sum_{(s,t)\in\mathcal{D}} \alpha_{s,t} K(x_{u,i}, x_{s,t})$.

then balance the $J$ functions with maximal $|\alpha_{s,t}|$, where $J$ is a hyper-parameter. This method aims to balance the kernel functions that contribute most to $e_{u,i}$, which leads to the following optimization

$$\min_{\theta_w} \sum_{(u,i)\in\mathcal{O}} \hat{w}_{u,i} \log \hat{w}_{u,i} + \gamma \sum_{j=1}^{J} \xi_j$$

$$\text{s.t.} \quad \xi_j \geq 0 \qquad j \in \{1,\ldots,J\} \quad \text{and} \quad \hat{w}_{u,i} \geq 0 \qquad \forall (u,i) \in \mathcal{O},$$

$$\sum_{(u,i)\in\mathcal{D}} o_{u,i}\hat{w}_{u,i} = 1,$$

$$\sum_{(u,i)\in\mathcal{D}} o_{u,i}\hat{w}_{u,i}h^{(j)}(x_{u,i}) - \frac{1}{|\mathcal{D}|}\sum_{(u,i)\in\mathcal{D}} h^{(j)}(x_{u,i}) \leq C + \xi_j \qquad j \in \{1,\ldots,J\},$$

$$\sum_{(u,i)\in\mathcal{D}} o_{u,i}\hat{w}_{u,i}h^{(j)}(x_{u,i}) - \frac{1}{|\mathcal{D}|}\sum_{(u,i)\in\mathcal{D}} h^{(j)}(x_{u,i}) \geq -C + \xi_j \qquad j \in \{1,\ldots,J\}.$$

The above optimization problem is equivalent to the following

$$\min_{\theta_w} \mathcal{L}_{\mathrm{w}}(\theta_w) = \sum_{(u,i)\in\mathcal{O}} \hat{w}_{u,i} \log \hat{w}_{u,i} + \gamma \sum_{j=1}^{J} \left( [-C - \hat{\tau}^{(j)}]_+ + [\hat{\tau}^{(j)} - C]_+ \right), \qquad (5)$$

where

$$\hat{\tau}^{(j)} = \sum_{(u,i)\in\mathcal{D}} o_{u,i}\hat{w}_{u,i}h^{(j)}(x_{u,i}) - \frac{1}{|\mathcal{D}|}\sum_{(u,i)\in\mathcal{D}} h^{(j)}(x_{u,i}) \qquad j \in \{1,\ldots,J\}.$$

Since achieving strict balancing constraints on all balancing functions is usually infeasible as $J$ increases, we introduce a slack variable $\xi_j$ and a pre-specified threshold $C$, which penalizes the loss when the deviation $|\hat{\tau}^j| > C$.

## 4.4 LEARNING ALGORITHM AND GENERALIZATION ERROR BOUNDS

Taking the AKBDR method as an example, because the balancing weights $\hat{w}_{u,i}$ and prediction errors $e_{u,i}$ are relying on each other, thus we adopt a widely used joint learning framework to train the prediction model $\hat{r}_{u,i} = f(x_{u,i}; \theta_r)$, balancing weight model $\hat{w}_{u,i} = g(x_{u,i}; \theta_w)$, and imputation model $\hat{e}_{u,i} = m(x_{u,i}; \theta_e)$ alternatively. Specifically, we train the prediction model by minimizing the $\mathcal{L}_{\mathrm{KBDR}}(\theta_r)$ loss shown in Equation 3, train the balancing weight model by minimizing the $\mathcal{L}_{\mathrm{w}}(\theta_w)$ in Equation 5, and train the imputation model by minimizing the loss function $\mathcal{L}_{\mathrm{e}}(\theta_e)$ below

$$\mathcal{L}_{\mathrm{e}}(\theta_e) = \frac{1}{|\mathcal{D}|} \sum_{(u,i)\in\mathcal{D}} o_{u,i}\hat{w}_{u,i}(\hat{e}_{u,i} - e_{u,i})^2, \qquad (6)$$

and the whole procedure of the proposed joint learning process is summarized in Alg. 1.

Next, we analyze the generalization bound of the KBIPS and KBDR methods.

**Theorem 2** (Generalization Bounds in RKHS). *Let $K$ be a bounded kernel, $\sup_x \sqrt{K(x,x)} = B < \infty$, and $B_K(M) = \{f \in \mathcal{F} \mid \|f\|_{\mathcal{F}} \leq M\}$ is the corresponding kernel-based hypotheses space. Suppose $\hat{w}_{u,i} \leq C$, $\delta(r, \cdot)$ is $L$-Lipschitz continuous for all $r$, and that $E_0 := \sup_r \delta(r, 0) < \infty$. Then with probability at least $1 - \eta$, we have*

$$\mathcal{L}_{\mathrm{Ideal}}(\theta_r) \leq \mathcal{L}_{\mathrm{KBIPS}}(\theta_r) + |\mathrm{Bias}(\mathcal{L}_{\mathrm{KBIPS}}(\theta_r))| + \frac{2CLMB}{\sqrt{|\mathcal{D}|}} + 5C(E_0 + LMB)\sqrt{\frac{\log(4/\eta)}{2|\mathcal{D}|}},$$

$$\mathcal{L}_{\mathrm{Ideal}}(\theta_r) \leq \mathcal{L}_{\mathrm{KBDR}}(\theta_r) + |\mathrm{Bias}(\mathcal{L}_{\mathrm{KBDR}}(\theta_r))| + (1 + 2C)\left( \frac{2LMB}{\sqrt{|\mathcal{D}|}} + 5(E_0 + LMB)\sqrt{\frac{\log(4/\eta)}{2|\mathcal{D}|}} \right).$$

Remarkably, the above generalization bounds in RKHS can be greatly reduced by adopting the proposed KBIPS and KBDR learning methods, because the prediction model minimizes the debiased losses $\mathcal{L}_{\mathrm{KBIPS}}(\theta_r)$ and $\mathcal{L}_{\mathrm{KBDR}}(\theta_r)$ during the model training phase, and $\mathrm{Bias}(\mathcal{L}_{\mathrm{KBIPS}}(\theta_r))$ and $\mathrm{Bias}(\mathcal{L}_{\mathrm{KBDR}}(\theta_r))$ can also be controlled via WKB or AKB methods.

---

**Algorithm 1:** The Proposed Adaptive KBDR (AKBDR) Learning Algorithm

---

**Input:** observed ratings $\mathbf{Y}^o$, and number of balancing functions $J$.

1 **while** *stopping criteria is not satisfied* **do**
2      **for** *number of steps for training the imputation model* **do**
3          Sample a batch of user-item pairs $\{(u_l, i_l)\}_{l=1}^{L}$ from $\mathcal{O}$;
4          Update $\theta_e$ by descending along the gradient $\nabla_{\theta_e} \mathcal{L}_e(\theta_e)$;
5      **end**
6      **for** *number of steps for training the balancing weight model* **do**
7          Sample a batch of user-item pairs $\{(u_m, i_m)\}_{m=1}^{M}$ from $\mathcal{D}$;
8          Solve the Equation 4 and select $J$ functions $h(x_{u_m,i_m})$ with maximum $|\alpha_{u_m,i_m}|$;
9          Update $\theta_w$ by descending along the gradient $\nabla_{\theta_w} \mathcal{L}_w(\theta_w)$;
10      **end**
11      **for** *number of steps for training the prediction model* **do**
12          Sample a batch of user-item pairs $\{(u_n, i_n)\}_{n=1}^{N}$ from $\mathcal{D}$;
13          Update $\theta_r$ by descending along the gradient $\nabla_{\theta_r} \mathcal{L}_{\text{KBDR}}(\theta_r)$;
14      **end**
15 **end**

---

## 5   RELATED WORK

**Debiased Collaborative Filtering.** Collaborative filtering (CF) plays an important role in today's digital and informative world (Chen et al., 2018; Huang et al., 2023; Lv et al., 2023; 2024). However, the collected data is observational rather than experimental, leading to various biases in the data, which seriously affects the quality of the learned model. One of the most important biases is the selection bias, which causes the distribution of the training data to be different from the distribution of the test data, thus making it challenging to achieve unbiased estimation and learning (Wang et al., 2022b; 2023b; Zou et al., 2023; Wang et al., 2023a; 2024). If we learn the model directly on the training data without debiasing, it will harm the prediction performance on the test data (Wang et al., 2023c; Zhang et al., 2023; Bai et al., 2024; Zhang et al., 2024). Many previous methods are proposed to mitigate the selection bias problem (Schnabel et al., 2016; Wang et al., 2019; Chen et al., 2021; Li et al., 2023c). The error-imputation-based (EIB) methods attempt to impute the missing events, and then train a CF model on both observed and imputed data (Chang et al., 2010; Steck, 2010; Hernández-Lobato et al., 2014). Another common type of debiasing method is propensity-based, including inverse propensity scoring (IPS) methods (Imbens and Rubin, 2015; Schnabel et al., 2016; Saito et al., 2020; Luo et al., 2021; Oosterhuis, 2022), and doubly robust (DR) methods (Morgan and Winship, 2015; Wang et al., 2019; Saito, 2020). Specifically, IPS adjusts the distribution by reweighting the observed events, while DR combines the EIB and IPS methods, which takes advantage of both, *i.e.*, has lower variance and bias. Based on the above advantages, many competing DR-based methods are proposed, such as MRDR (Guo et al., 2021), DR-BIAS (Dai et al., 2022), ESCM$^2$-DR (Wang et al., 2022a), TDR (Li et al., 2023b), SDR (Li et al., 2023f), and N-DR (Li et al., 2024). Given the widespread of the propensity model, Li et al. (2023d) proposed a method to train balancing weights with a few unbiased ratings for debiasing. More recently, Li et al. (2023e) proposed a propensity balancing measurement to regularize the IPS and DR estimators. In this paper, we extend the above idea by proposing novel kernel-based balancing IPS and DR estimators that adaptively find the balancing functions that contribute the most to reducing the estimation bias.

**Covariate Balancing in Causal Inference.** Balancing refers to aligning the distribution of covariates in the treatment and control groups, which is crucial to the estimation of causal effects based on observational datasets (Stuart, 2010; Imbens and Rubin, 2015). This is because balancing ensures that units receiving different treatments are comparable directly, and the association becomes causation under the unconfoundedness assumption (Imai and Ratkovic, 2014; Hernán and Robins, 2020). In randomized controlled experiments, balancing is naturally maintained due to the complete random assignment of treatments. However, in observational studies, treatment groups typically exhibit systematic differences in covariates, which can result in a lack of balance. To obtain accurate estimates of causal effects in observational studies, a wide variety of methods have emerged for balancing the finite order moments of covariates, including matching (Rosenbaum and Rubin, 1983; Stuart, 2010;

Table 1: Performance on AUC, NDCG@$K$, and F1@$K$ on COAT, MUSIC and PRODUCT. The best two results are bolded and the best baseline result is underlined for IPS-based and DR-based methods.

| Method | COAT | | | MUSIC | | | PRODUCT | | |
|---|---|---|---|---|---|---|---|---|---|
| | AUC | NDCG@5 | F1@5 | AUC | NDCG@5 | F1@5 | AUC | NDCG@20 | F1@20 |
| MF | $0.703_{\pm0.006}$ | $0.605_{\pm0.012}$ | $0.467_{\pm0.007}$ | $0.673_{\pm0.001}$ | $0.635_{\pm0.002}$ | $0.306_{\pm0.002}$ | $0.753_{\pm0.001}$ | $0.449_{\pm0.002}$ | $0.124_{\pm0.002}$ |
| + IPS | $0.717_{\pm0.007}$ | $0.617_{\pm0.009}$ | $0.473_{\pm0.008}$ | $0.678_{\pm0.001}$ | $0.638_{\pm0.002}$ | $0.318_{\pm0.002}$ | $0.755_{\pm0.004}$ | $0.452_{\pm0.010}$ | $0.131_{\pm0.004}$ |
| + SNIPS | $0.714_{\pm0.012}$ | $0.614_{\pm0.012}$ | $0.474_{\pm0.009}$ | $0.683_{\pm0.002}$ | $0.639_{\pm0.002}$ | $0.316_{\pm0.002}$ | $0.754_{\pm0.003}$ | $0.453_{\pm0.004}$ | $0.126_{\pm0.003}$ |
| + ASIPS | $0.719_{\pm0.009}$ | $0.618_{\pm0.012}$ | $0.476_{\pm0.009}$ | $0.679_{\pm0.003}$ | $0.640_{\pm0.003}$ | $0.319_{\pm0.003}$ | $0.757_{\pm0.005}$ | $0.474_{\pm0.007}$ | $0.130_{\pm0.005}$ |
| + IPS-V2 | $\underline{0.726}_{\pm0.005}$ | $\underline{0.627}_{\pm0.009}$ | $\underline{0.479}_{\pm0.005}$ | $\underline{0.685}_{\pm0.002}$ | $\underline{0.646}_{\pm0.003}$ | $\underline{0.320}_{\pm0.002}$ | $\underline{0.764}_{\pm0.001}$ | $\underline{0.476}_{\pm0.003}$ | $\underline{0.135}_{\pm0.003}$ |
| + RKBIPS-Exp | $0.714_{\pm0.003}$ | $0.618_{\pm0.010}$ | $0.474_{\pm0.007}$ | $0.676_{\pm0.002}$ | $0.642_{\pm0.003}$ | $0.318_{\pm0.002}$ | $0.763_{\pm0.001}$ | $0.463_{\pm0.007}$ | $0.134_{\pm0.002}$ |
| + RKBIPS-Gau | $0.715_{\pm0.005}$ | $0.619_{\pm0.010}$ | $0.475_{\pm0.008}$ | $0.678_{\pm0.001}$ | $0.640_{\pm0.004}$ | $0.315_{\pm0.003}$ | $0.760_{\pm0.003}$ | $0.470_{\pm0.008}$ | $0.133_{\pm0.003}$ |
| + WKBIPS-Exp | $0.723_{\pm0.004}$ | $0.624_{\pm0.009}$ | $0.480_{\pm0.007}$ | $0.687_{\pm0.002}$ | $0.654_{\pm0.002}$ | $0.322_{\pm0.002}$ | $0.765_{\pm0.003}$ | $0.475_{\pm0.007}$ | $0.138_{\pm0.003}$ |
| + WKBIPS-Gau | $0.722_{\pm0.004}$ | $0.625_{\pm0.008}$ | $0.479_{\pm0.007}$ | $0.686_{\pm0.002}$ | $0.650_{\pm0.002}$ | $0.321_{\pm0.002}$ | $0.763_{\pm0.003}$ | $0.476_{\pm0.007}$ | $0.137_{\pm0.003}$ |
| + AKBIPS-Exp | $\mathbf{0.732}^{*}_{\pm0.004}$ | $\mathbf{0.636}^{*}_{\pm0.006}$ | $\mathbf{0.483}^{*}_{\pm0.006}$ | $\mathbf{0.689}^{*}_{\pm0.001}$ | $\mathbf{0.658}^{*}_{\pm0.002}$ | $\mathbf{0.324}^{*}_{\pm0.002}$ | $\mathbf{0.766}^{*}_{\pm0.003}$ | $\mathbf{0.478}^{*}_{\pm0.009}$ | $\mathbf{0.138}^{*}_{\pm0.003}$ |
| + AKBIPS-Gau | $\mathbf{0.730}^{*}_{\pm0.003}$ | $\mathbf{0.633}^{*}_{\pm0.008}$ | $\mathbf{0.484}_{\pm0.007}$ | $\mathbf{0.688}^{*}_{\pm0.003}$ | $\mathbf{0.655}^{*}_{\pm0.003}$ | $\mathbf{0.324}^{*}_{\pm0.002}$ | $\mathbf{0.767}^{*}_{\pm0.003}$ | $\mathbf{0.480}_{\pm0.009}$ | $\mathbf{0.139}^{*}_{\pm0.003}$ |
| + DR | $0.718_{\pm0.008}$ | $0.623_{\pm0.009}$ | $0.474_{\pm0.007}$ | $0.684_{\pm0.002}$ | $0.658_{\pm0.003}$ | $0.326_{\pm0.002}$ | $0.755_{\pm0.003}$ | $0.462_{\pm0.010}$ | $0.135_{\pm0.005}$ |
| + DR-JL | $0.723_{\pm0.005}$ | $0.629_{\pm0.007}$ | $0.479_{\pm0.005}$ | $0.685_{\pm0.002}$ | $0.653_{\pm0.003}$ | $0.324_{\pm0.002}$ | $0.766_{\pm0.005}$ | $0.467_{\pm0.005}$ | $0.136_{\pm0.003}$ |
| + MRDR-JL | $0.727_{\pm0.005}$ | $0.627_{\pm0.008}$ | $0.480_{\pm0.008}$ | $0.684_{\pm0.002}$ | $0.652_{\pm0.003}$ | $0.325_{\pm0.002}$ | $0.768_{\pm0.005}$ | $0.473_{\pm0.007}$ | $0.139_{\pm0.004}$ |
| + DR-BIAS | $0.726_{\pm0.004}$ | $0.629_{\pm0.009}$ | $0.482_{\pm0.007}$ | $0.685_{\pm0.002}$ | $0.653_{\pm0.002}$ | $0.325_{\pm0.003}$ | $0.768_{\pm0.003}$ | $0.477_{\pm0.006}$ | $0.137_{\pm0.004}$ |
| + DR-MSE | $0.727_{\pm0.007}$ | $0.631_{\pm0.008}$ | $0.484_{\pm0.007}$ | $0.687_{\pm0.002}$ | $0.657_{\pm0.003}$ | $0.327_{\pm0.002}$ | $0.770_{\pm0.002}$ | $0.480_{\pm0.006}$ | $0.140_{\pm0.003}$ |
| + MR | $0.724_{\pm0.004}$ | $0.636_{\pm0.006}$ | $0.481_{\pm0.006}$ | $\underline{0.691}_{\pm0.002}$ | $0.647_{\pm0.002}$ | $0.316_{\pm0.003}$ | $\underline{0.776}_{\pm0.005}$ | $0.483_{\pm0.006}$ | $0.142_{\pm0.003}$ |
| + TDR | $0.714_{\pm0.006}$ | $0.634_{\pm0.011}$ | $0.483_{\pm0.008}$ | $0.688_{\pm0.003}$ | $\mathbf{\underline{0.662}}_{\pm0.002}$ | $\mathbf{0.329}_{\pm0.002}$ | $0.772_{\pm0.003}$ | $0.486_{\pm0.005}$ | $0.140_{\pm0.003}$ |
| + TDR-JL | $0.731_{\pm0.007}$ | $0.639_{\pm0.007}$ | $0.484_{\pm0.007}$ | $0.689_{\pm0.002}$ | $0.656_{\pm0.004}$ | $0.327_{\pm0.002}$ | $0.772_{\pm0.003}$ | $0.489_{\pm0.005}$ | $0.142_{\pm0.003}$ |
| + SDR | $\underline{0.735}_{\pm0.005}$ | $\underline{0.640}_{\pm0.007}$ | $0.484_{\pm0.006}$ | $0.688_{\pm0.002}$ | $0.661_{\pm0.003}$ | $\mathbf{\underline{0.329}}_{\pm0.002}$ | $0.773_{\pm0.001}$ | $\underline{0.491}_{\pm0.003}$ | $\underline{0.143}_{\pm0.003}$ |
| + DR-V2 | $0.734_{\pm0.007}$ | $0.639_{\pm0.009}$ | $\underline{0.487}_{\pm0.006}$ | $0.690_{\pm0.002}$ | $0.660_{\pm0.005}$ | $0.328_{\pm0.002}$ | $0.773_{\pm0.003}$ | $0.488_{\pm0.006}$ | $0.142_{\pm0.004}$ |
| + RKBDR-Exp | $0.730_{\pm0.003}$ | $0.631_{\pm0.005}$ | $0.482_{\pm0.006}$ | $0.682_{\pm0.002}$ | $0.648_{\pm0.003}$ | $0.323_{\pm0.002}$ | $0.765_{\pm0.004}$ | $0.460_{\pm0.006}$ | $0.138_{\pm0.003}$ |
| + RKBDR-Gau | $0.726_{\pm0.005}$ | $0.630_{\pm0.008}$ | $0.480_{\pm0.008}$ | $0.683_{\pm0.002}$ | $0.652_{\pm0.003}$ | $0.325_{\pm0.002}$ | $0.766_{\pm0.003}$ | $0.469_{\pm0.007}$ | $0.134_{\pm0.004}$ |
| + WKBDR-Exp | $0.735_{\pm0.005}$ | $0.637_{\pm0.009}$ | $0.483_{\pm0.006}$ | $0.685_{\pm0.003}$ | $0.654_{\pm0.003}$ | $0.325_{\pm0.002}$ | $0.773_{\pm0.003}$ | $0.489_{\pm0.008}$ | $0.142_{\pm0.003}$ |
| + WKBDR-Gau | $0.732_{\pm0.003}$ | $0.638_{\pm0.007}$ | $0.483_{\pm0.005}$ | $0.687_{\pm0.002}$ | $0.655_{\pm0.002}$ | $0.327_{\pm0.002}$ | $0.773_{\pm0.002}$ | $0.490_{\pm0.005}$ | $0.142_{\pm0.004}$ |
| + AKBDR-Exp | $\mathbf{0.745}^{*}_{\pm0.004}$ | $\mathbf{0.645}^{*}_{\pm0.008}$ | $\mathbf{0.493}^{*}_{\pm0.007}$ | $\mathbf{0.692}_{\pm0.002}$ | $0.661_{\pm0.002}$ | $0.328_{\pm0.002}$ | $\mathbf{0.782}^{*}_{\pm0.003}$ | $\mathbf{0.498}^{*}_{\pm0.008}$ | $\mathbf{0.147}^{*}_{\pm0.003}$ |
| + AKBDR-Gau | $\mathbf{0.746}^{*}_{\pm0.004}$ | $\mathbf{0.646}^{*}_{\pm0.008}$ | $\mathbf{0.492}_{\pm0.007}$ | $\mathbf{0.694}^{*}_{\pm0.002}$ | $\mathbf{0.664}^{*}_{\pm0.002}$ | $\mathbf{0.332}^{*}_{\pm0.002}$ | $\mathbf{0.782}^{*}_{\pm0.005}$ | $\mathbf{0.503}^{*}_{\pm0.006}$ | $\mathbf{0.148}^{*}_{\pm0.004}$ |

Note: * means statistically significant results (p-value $\leq 0.05$) using the paired-t-test compared with the best baseline method.

Wu et al., 2020), stratification (Hernán and Robins, 2020), entropy balancing (Hainmueller, 2012; Zhao and Percival, 2017), covariate balancing (Imai and Ratkovic, 2014), and weighted euclidean balancing (Chen and Zhou, 2023). In recent years, several approaches are developed balancing infinite order moments of covariates (Sant'Anna et al., 2022) or the covariates distributions (Wong and Chan, 2018). However, it is unrealistic to balance infinite order moments with only finite samples, therefore in this paper we propose a novel balancing method that adaptively finds the balancing functions in RKHS that are most important for achieving unbiased learning.

## 6 EXPERIMENTS

**Datasets and Experimental Details.** Following the previous studies (Wang et al., 2019; Chen et al., 2021; Wang et al., 2021; Li et al., 2023b), we conduct real-world experiments on three widely used benchmark datasets: COAT, MUSIC, and a large-scale industrial dataset PRODUCT. The COAT dataset consists of 6,960 biased ratings and 4,640 unbiased ratings evaluated by 290 users to 300 items. The MUSIC dataset consists of 311,704 biased ratings and 54,000 unbiased ratings evaluated by 15,400 users to 1,000 items. The PRODUCT dataset consists of 4,676,570 records of video watching ratios from 1,411 users to 3,327 items and is almost fully exposed. Both COAT and MUSIC are five-scale datasets, and we binarize the ratings less than three as 0, otherwise as 1. For the PRODUCT dataset, we binarize the video watching ratios less than two as 0, otherwise as 1. We adopt three widely used evaluation metrics: AUC, NDCG@$K$, and F1@$K$ to measure the debiasing performance. We set $K = 5$ for COAT and MUSIC and $K = 20$ for PRODUCT. We use Adam as the optimizer and tune the learning rate in $\{0.01, 0.03, 0.05, 0.1\}$, weight decay in $[1e-6, 5e-3]$, margin threshold $C$ in $\{1e-6, 5e-5, 1e-5, \ldots, 1\}$, kernel hyper-parameter $\sigma^2$ in $\{0.5, 1, 5\}$ for both Gaussian and exponential kernels, and regularization hyper-parameter $\gamma$ in $\{1, 2, 5, 10, 20, 50\}$. We set the batch size to 128 on COAT and 2,048 on MUSIC and PRODUCT.

**Baselines.** We implement our proposed RKB, WKB, and AKB methods with both Gaussian and exponential kernels by taking **MF** (Koren et al., 2009) as the base model. We compare our methods with the following IPS-based baselines: **IPS** (Schnabel et al., 2016), **SNIPS** (Schnabel et al., 2016), **ASIPS** (Saito, 2020), and **IPS-V2** (Li et al., 2023e). Meanwhile, we also compare our methods with the following DR-based baselines: **DR** (Saito, 2020), **DR-JL** (Wang et al., 2019), **MRDR** (Guo et al., 2021), **DR-BIAS** (Dai et al., 2022), **DR-MSE** (Dai et al., 2022), **MR** (Li et al., 2023a), **TDR** (Li et al., 2023b), **TDR-JL** (Li et al., 2023b), **StableDR** (Li et al., 2023f), and **DR-V2** (Li et al., 2023e).

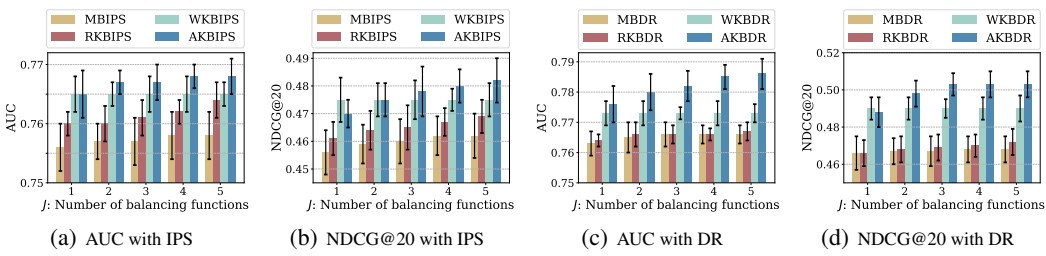

Figure 1: Effects of the value of $J$ on AUC and NDCG@20 on the PRODUCT dataset.

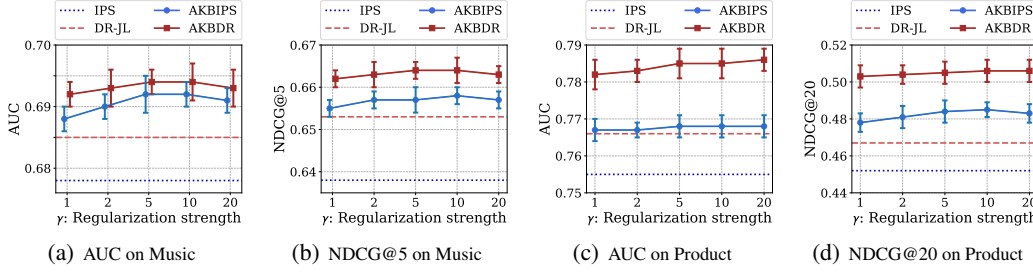

Figure 2: Effects of hyper-parameter $\gamma$ on AUC and NDCG@$K$ on MUSIC and PRODUCT datasets.

**Performance Comparison.** We compare the proposed methods with previous methods, as shown in Table 1. First, all the debiasing methods outperform MF, and AKBDR achieves the optimal performance on all three datasets with either Gaussian or exponential kernels. Second, methods with balancing properties such as IPS-V2 and DR-V2 achieve competitive performance, which demonstrates the importance of propensity balancing. Third, for our methods, RKB methods perform the worst due to the insufficient balancing caused by random selected functions, while AKB methods perform the best due to the most important kernel functions are adaptively found and balanced.

**In-Depth Analysis.** We further explore the impact of the value of $J$ on the debiasing performance of kernel balancing methods on the PRODUCT dataset. We also implement the moment balancing (MB) methods which balance the first $J$-th order moments for comparison and the results are shown in Figure 1. We find the performance for all methods except WKB increases monotonically as the value of $J$ increases because more functions or moments being balanced leads to less bias. Since the WKB methods focus on controlling the worst-case, the performance does not change for different $J$ and shows competitive performance with AKB methods when $J$ is small (*e.g.*, $J = 1$). In addition, kernel balancing methods stably outperform moment balancing methods with varying values of $J$ even if the balancing functions are selected randomly, validating the effectiveness of kernel balancing.

**Sensitivity Analysis.** To explore the effect of balancing regularization hyper-parameter $\gamma$ on debiasing performance, we conduct sensitivity analysis on AKB methods with varying $\gamma$ in $\{1, 2, 5, 10, 20\}$ on the MUSIC and PRODUCT datasets, as shown in Figure 2. The AKB methods consistently outperform the baseline methods under different regularization strengths. Specifically, even when the balancing constraint strength is small, the AKB method can still get significant performance gains, and the optimal performance is achieved around the moderate $\gamma$ (*e.g.*, 5 or 10).

## 7 CONCLUSION

In the information-driven landscape, collaborative filtering is pivotal for various e-commerce platforms. However, selection bias in the collected data poses a great challenge for collaborative filtering model training. To mitigate this issue, this paper theoretically reveal that previous debiased collaborative filtering approaches are restricted to balancing finite-dimensional pre-specified functions of features. To fill the gap, we first develop two new estimators, KBIPS and KBDR, which extend the widely-used IPS and DR estimators in debiased collaborative filtering. Then we propose a universal kernel-based balancing method that adaptively achieves balance for the selected functions in an RKHS. Based on it, we further propose an adaptive kernel balancing method. Theoretical analysis demonstrates that the proposed balancing method reduces both estimation bias and the generalization bound. Extensive experiments on real-world datasets validate the effectiveness of our methods.

## ACKNOWLEDGEMENT

This work was supported in part by National Natural Science Foundation of China (No. 623B2002, 62102420, 12301370)

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

## A PROOFS

**Theorem 1.** *If $e_{u,i} \in \mathcal{H}_J = \mathrm{span}\{h^{(1)}(\cdot), \ldots, h^{(J)}(\cdot)\}$, then the above learned propensities lead to an unbiased ideal loss estimation in term of the IPS method.*

*Proof.* The balancing weight $\hat{w}_{u,i}$ is learned by solving the following optimization problem

$$\max_{\theta_w} - \sum_{(u,i) \in \mathcal{O}} \hat{w}_{u,i} \log \hat{w}_{u,i}$$

$$\text{s.t.} \quad \frac{1}{|\mathcal{D}|} \sum_{(u,i) \in \mathcal{D}} o_{u,i} \hat{w}_{u,i} h^{(j)}(x_{u,i}) = \frac{1}{|\mathcal{D}|} \sum_{(u,i) \in \mathcal{D}} h^{(j)}(x_{u,i}) \qquad j \in \{1, \ldots, J\},$$

$$\frac{1}{|\mathcal{D}|} \sum_{(u,i) \in \mathcal{D}} o_{u,i} \hat{w}_{u,i} = 1, \qquad \hat{w}_{u,i} \geq 0 \qquad \forall (u,i) \in \mathcal{O},$$

thus the learned $\hat{w}_{u,i}$ satisfying the constraints in the optimization problem, which means that

$$\frac{1}{|\mathcal{D}|} \sum_{(u,i) \in \mathcal{D}} (o_{u,i} \hat{w}_{u,i} h^{(j)}(x_{u,i}) - h^{(j)}(x_{u,i})) = 0, \quad j \in \{1, \ldots, J\}.$$

If $e_{u,i} \in \mathcal{H}_J = \mathrm{span}\{h^{(1)}(\cdot), \ldots, h^{(J)}(\cdot)\}$, there exist $\{\alpha_j\}_{j=1}^J$ satisfying $e_{u,i} = \sum_{j=1}^J \alpha_j h^{(j)}(x_{u,i})$. the empirical bias of the KBIPS estimator for estimating the ideal loss is

$$\mathrm{Bias}(\mathcal{L}_{\mathrm{KBIPS}}(\theta_r)) = \left\{ \frac{1}{|\mathcal{D}|} \sum_{(u,i) \in \mathcal{D}} (o_{u,i} \hat{w}_{u,i} - 1) e_{u,i} \right\}^2$$

$$= \left\{ \frac{1}{|\mathcal{D}|} \sum_{(u,i) \in \mathcal{D}} (o_{u,i} \hat{w}_{u,i} - 1) \sum_{j=1}^J \alpha_j h^{(j)}(x_{u,i}) \right\}^2$$

$$= \left\{ \frac{1}{|\mathcal{D}|} \sum_{j=1}^J \alpha_j \sum_{(u,i) \in \mathcal{D}} (o_{u,i} \hat{w}_{u,i} h^{(j)}(x_{u,i}) - h^{(j)}(x_{u,i})) \right\}^2$$

$$= 0.$$

If $e_{u,i} \notin \mathcal{H}_J = \mathrm{span}\{h^{(1)}(\cdot), \ldots, h^{(J)}(\cdot)\}$, then for all $\{\alpha_j\}_{j=1}^J$, $e_{u,i} = \sum_{j=1}^J \alpha_j h^{(j)}(x) + \epsilon(x_{u,i})$, where $\epsilon(x_{u,i})$ is the non-zero residual term. Therefore, we have

$$\mathrm{Bias}(\mathcal{L}_{\mathrm{KBIPS}}(\theta_r)) = \left\{ \frac{1}{|\mathcal{D}|} \sum_{(u,i) \in \mathcal{D}} (o_{u,i} \hat{w}_{u,i} - 1) e_{u,i} \right\}^2$$

$$= \left\{ \frac{1}{|\mathcal{D}|} \sum_{(u,i) \in \mathcal{D}} (o_{u,i} \hat{w}_{u,i} - 1) \epsilon(x_{u,i}) \right\}^2 \neq 0.$$

A similar argument also holds for the proposed KBDR estimator. $\qquad \square$

**Lemma 1** (Sriperumbudur et al. (2011)). *Both the Gaussian and exponential kernels are universal.*

*Proof.* The proof can be found in Sriperumbudur et al. (2011). $\qquad \square$

**Lemma 2** (Representer theorem). *If $\Omega = h(\|f\|)$ for some increasing function $h : \mathbb{R}_+ \to \overline{\mathbb{R}}$, then some empirical risk minimizer must admit the form $f(\cdot) = \sum_{i=1}^n \alpha_i K(\cdot, x_i)$ for some $\boldsymbol{\alpha} = (\alpha_1, \ldots, \alpha_n) \in \mathbb{R}^n$. If $h$ is strictly increasing, all minimizers admit this form.*

*Proof.* The proof can be found in Theorem 6.11 of Mohri et al. (2018). □

**Definition 2** (Empirical Rademacher Complexity (Shalev-Shwartz and Ben-David, 2014))**.** *Let $\mathcal{F}$ be a family of prediction models mapping from $x \in \mathcal{X}$ to $[a, b]$, and $S = \{x_{u,i} \mid (u, i) \in \mathcal{D}\}$ a fixed sample of size $|\mathcal{D}|$ with elements in $\mathcal{X}$. Then, the empirical Rademacher complexity of $\mathcal{F}$ with respect to the sample $S$ is defined as*

$$\mathcal{R}(\mathcal{F}) = \mathbb{E}_{\boldsymbol{\sigma} \sim \{-1, +1\}^{|\mathcal{D}|}} \sup_{f \in \mathcal{F}} \left[ \frac{1}{|\mathcal{D}|} \sum_{(u,i) \in \mathcal{D}} \sigma_{u,i} f(x_{u,i}) \right],$$

*where $\boldsymbol{\sigma} = \{\sigma_{u,i} : (u, i) \in \mathcal{D}\}$, and $\sigma_{u,i}$ are independent uniform random variables taking values in $\{-1, +1\}$. The random variables $\sigma_{u,i}$ are called Rademacher variables.*

**Lemma 3** (Rademacher Comparison Lemma (Shalev-Shwartz and Ben-David, 2014))**.** *Let $\mathcal{F}$ be a family of real-valued functions on $z \in \mathcal{Z}$ to $[a, b]$, and $S = \{x_{u,i} \mid (u, i) \in \mathcal{D}\}$ a fixed sample of size $|\mathcal{D}|$ with elements in $\mathcal{X}$. Then*

$$\mathbb{E}_{S \sim \mathbb{P}^{|\mathcal{D}|}} \left[ \sup_{f \in \mathcal{F}} \left( \mathbb{E}_{z \sim \mathbb{P}}[f(z)] - \frac{1}{|\mathcal{D}|} \sum_{(u,i) \in \mathcal{D}} f(z_{u,i}) \right) \right] \le 2 \mathbb{E}_{S \sim \mathbb{P}^{|\mathcal{D}|}} \mathbb{E}_{\boldsymbol{\sigma} \sim \{-1, +1\}^{|\mathcal{D}|}} \sup_{f \in \mathcal{F}} \left[ \frac{1}{|\mathcal{D}|} \sum_{(u,i) \in \mathcal{D}} \sigma_{u,i} f(z_{u,i}) \right],$$

*where $\boldsymbol{\sigma} = \{\sigma_{u,i} : (u, i) \in \mathcal{D}\}$, and $\sigma_{u,i}$ are independent uniform random variables taking values in $\{-1, +1\}$. The random variables $\sigma_{u,i}$ are called Rademacher variables.*

*Proof.* The proof can be found in Lemma 26.2 of Shalev-Shwartz and Ben-David (2014). □

**Lemma 4** (McDiarmid's Inequality (Shalev-Shwartz and Ben-David, 2014))**.** *Let $V$ be some set and let $f : V^m \to \mathbb{R}$ be a function of $m$ variables such that for some $c > 0$, for all $i \in [m]$ and for all $x_1, \ldots, x_m, x_i' \in V$ we have*

$$|f(x_1, \ldots, x_m) - f(x_1, \ldots, x_{i-1}, x_i', x_{i+1}, \ldots, x_m)| \le c.$$

*Let $X_1, \ldots, X_m$ be $m$ independent random variables taking values in $V$. Then, with the probability of at least $1 - \delta$ we have*

$$|f(X_1, \ldots, X_m) - \mathbb{E}[f(X_1, \ldots, X_m)]| \le c \sqrt{\log\left(\frac{2}{\delta}\right) m/2}.$$

*Proof.* The proof can be found in Lemma 26.4 of Shalev-Shwartz and Ben-David (2014). □

**Lemma 5** (Rademacher Calculus (Shalev-Shwartz and Ben-David, 2014))**.** *For any $A \subset \mathbb{R}^m$, scalar $c \in \mathbb{R}$, and vector $\mathbf{a}_0 \in \mathbb{R}^m$, we have*

$$\mathcal{R}(\{c\mathbf{a} + \mathbf{a}_0 : \mathbf{a} \in A\}) \le |c| \mathcal{R}(A).$$

*Proof.* The proof can be found in Lemma 26.6 of Shalev-Shwartz and Ben-David (2014). □

**Lemma 6** (Talagrand's Lemma (Mohri et al., 2018))**.** *Let $\Phi_1, \ldots, \Phi_m$ be $L$-Lipschitz functions from $\mathbb{R}$ to $\mathbb{R}$ and $\sigma_1, \ldots, \sigma_m$ be Rademacher random variables. Then, for any hypothesis set $\mathcal{F}$ of real-valued functions, the following inequality holds*

$$\frac{1}{m} \mathbb{E}_{\boldsymbol{\sigma}} \left[ \sup_{f \in \mathcal{F}} \sum_{i=1}^{m} \sigma_i (\Phi_i \circ f)(x_i) \right] \le \frac{L}{m} \mathbb{E}_{\boldsymbol{\sigma}} \left[ \sup_{f \in \mathcal{F}} \sum_{i=1}^{m} \sigma_i f(x_i) \right] = L\mathcal{R}(\mathcal{F}).$$

*In particular, if $\Phi_i = \Phi$ for all $i \in [m]$, then the following holds*

$$\mathcal{R}(\Phi \circ \mathcal{F}) \le L\mathcal{R}(\mathcal{F}).$$

*Proof.* The proof can be found in Section 5.4 of Mohri et al. (2018). □

**Lemma 7.** *Suppose $K$ is a bounded kernel with $\sup_x \sqrt{K(x,x)} = B < \infty$ and let $\mathcal{F}$ be its RKHS. Let $M > 0$ be fixed. Then for any $S = \{x_{u,i} : (u,i) \in \mathcal{D}\}$,*

$$\mathcal{R}\left(B_K(M)\right) \leq \frac{MB}{\sqrt{|\mathcal{D}|}},$$

*where $B_K(M) = \{f \in \mathcal{F} \mid \|f\|_{\mathcal{F}} \leq M\}$.*

*Proof.* Fix $S = \{x_{u,i} : (u,i) \in \mathcal{D}\}$. Then

$$\mathcal{R}\left(B_K(M)\right) = \mathbb{E}_\sigma \left[ \sup_{f \in B_K(M)} \frac{1}{|\mathcal{D}|} \sum_{(u,i) \in \mathcal{D}} \sigma_{u,i} f\left(x_{u,i}\right) \right]$$

$$= \frac{1}{|\mathcal{D}|} \mathbb{E}_\sigma \left[ \sup_{f \in B_K(M)} \sum_{(u,i) \in \mathcal{D}} \sigma_{u,i} \left\langle f, K\left(\cdot, x_{u,i}\right) \right\rangle \right]$$

$$= \frac{1}{|\mathcal{D}|} \mathbb{E}_\sigma \left[ \sup_{f \in B_K(M)} \left\langle f, \sum_{(u,i) \in \mathcal{D}} \sigma_{u,i} K\left(\cdot, x_{u,i}\right) \right\rangle \right]$$

$$= \frac{1}{|\mathcal{D}|} \mathbb{E}_\sigma \left[ \left\langle M \frac{\sum_{(u,i) \in \mathcal{D}} \sigma_{u,i} K\left(\cdot, x_{u,i}\right)}{\left\| \sum_{(u,i) \in \mathcal{D}} \sigma_{u,i} K\left(\cdot, x_{u,i}\right) \right\|}, \sum_{(u,i) \in \mathcal{D}} \sigma_{u,i} K\left(\cdot, x_{u,i}\right) \right\rangle \right]$$

$$= \frac{M}{|\mathcal{D}|} \mathbb{E}_\sigma \left[ \left\| \sum_{(u,i) \in \mathcal{D}} \sigma_{u,i} K\left(\cdot, x_{u,i}\right) \right\| \right]$$

$$= \frac{M}{|\mathcal{D}|} \mathbb{E}_\sigma \left[ \sqrt{\left\| \sum_{(u,i) \in \mathcal{D}} \sigma_{u,i} K\left(\cdot, x_{u,i}\right) \right\|^2} \right]$$

$$\leq \frac{M}{|\mathcal{D}|} \sqrt{\mathbb{E}_\sigma \left\| \sum_{(u,i) \in \mathcal{D}} \sigma_{u,i} K\left(\cdot, x_{u,i}\right) \right\|^2}$$

$$= \frac{M}{|\mathcal{D}|} \sqrt{\sum_{(u,i) \in \mathcal{D}} \left\| K\left(\cdot, x_{u,i}\right) \right\|^2}$$

$$= \frac{M}{|\mathcal{D}|} \sqrt{\sum_{(u,i) \in \mathcal{D}} K\left(x_{u,i}, x_{u,i}\right)}$$

$$\leq \frac{M}{|\mathcal{D}|} \sqrt{|\mathcal{D}| B^2}$$

$$= \frac{MB}{\sqrt{|\mathcal{D}|}}.$$

$\square$

**Theorem 2** (Generalization Bounds in RKHS). *Let $K$ be a bounded kernel, $\sup_x \sqrt{K(x,x)} = B < \infty$, and $B_K(M) = \{f \in \mathcal{F} \mid \|f\|_{\mathcal{F}} \leq M\}$ is the corresponding kernel-based hypotheses space. Suppose $\hat{w}_{u,i} \leq C$, $\delta(r, \cdot)$ is $L$-Lipschitz continuous for all $r$, and that $E_0 := \sup_r \delta(r, 0) < \infty$. Then with probability at least $1 - \eta$, we have*

$$\mathcal{L}_{\text{Ideal}}(\theta_r) \leq \mathcal{L}_{\text{KBIPS}}(\theta_r) + |\text{Bias}(\mathcal{L}_{\text{KBIPS}}(\theta_r))| + \frac{2CLMB}{\sqrt{|\mathcal{D}|}} + 5C(E_0 + LMB)\sqrt{\frac{\log(4/\eta)}{2|\mathcal{D}|}},$$

*and*

$$\mathcal{L}_{\text{Ideal}}(\theta_r) \leq \mathcal{L}_{\text{KBDR}}(\theta_r) + |\text{Bias}(\mathcal{L}_{\text{KBDR}}(\theta_r))| + (1 + 2C)\left(\frac{2LMB}{\sqrt{|\mathcal{D}|}} + 5(E_0 + LMB)\sqrt{\frac{\log(4/\eta)}{2|\mathcal{D}|}}\right).$$

*Proof.* We first prove the generalization bound of the KBIPS estimator, noting that the ideal loss can be decomposed as

$$
\begin{aligned}
\mathcal{L}_{\text{Ideal}}(\theta_r) &= \mathcal{L}_{\text{KBIPS}}(\theta_r) + (\mathcal{L}_{\text{Ideal}}(\theta_r) - \mathbb{E}(\mathcal{L}_{\text{KBIPS}}(\theta_r))) + (\mathbb{E}(\mathcal{L}_{\text{KBIPS}}(\theta_r)) - \mathcal{L}_{\text{KBIPS}}(\theta_r)) \\
&= \mathcal{L}_{\text{KBIPS}}(\theta_r) + \text{Bias}(\mathcal{L}_{\text{KBIPS}}(\theta_r)) + (\mathbb{E}(\mathcal{L}_{\text{KBIPS}}(\theta_r)) - \mathcal{L}_{\text{KBIPS}}(\theta_r)) \\
&\leq \mathcal{L}_{\text{KBIPS}}(\theta_r) + |\text{Bias}(\mathcal{L}_{\text{KBIPS}}(\theta_r))| \\
&\quad + \sup_{f_\theta \in B_K(M)} \left(\mathbb{E}\left[\frac{1}{|\mathcal{D}|}\sum_{(u,i)\in\mathcal{D}} o_{u,i}\hat{w}_{u,i}e_{u,i}\right] - \frac{1}{|\mathcal{D}|}\sum_{(u,i)\in\mathcal{D}} o_{u,i}\hat{w}_{u,i}e_{u,i}\right).
\end{aligned}
$$

For simplicity, we denote the last term in the above formula as

$$\mathcal{B}(\mathcal{F}) = \sup_{f_\theta \in B_K(M)} \left(\mathbb{E}\left[\frac{1}{|\mathcal{D}|}\sum_{(u,i)\in\mathcal{D}} o_{u,i}\hat{w}_{u,i}e_{u,i}\right] - \frac{1}{|\mathcal{D}|}\sum_{(u,i)\in\mathcal{D}} o_{u,i}\hat{w}_{u,i}e_{u,i}\right),$$

we then aim to bound $\mathcal{B}(\mathcal{F})$ in the following.

Note that

$$\mathcal{B}(\mathcal{F}) = \mathop{\mathbb{E}}_{S\sim\mathbb{P}^{|\mathcal{D}|}}[\mathcal{B}(\mathcal{F})] + \left\{\mathcal{B}(\mathcal{F}) - \mathop{\mathbb{E}}_{S\sim\mathbb{P}^{|\mathcal{D}|}}[\mathcal{B}(\mathcal{F})]\right\},$$

where the first term is $\mathop{\mathbb{E}}_{S\sim\mathbb{P}^{|\mathcal{D}|}}[\mathcal{B}(\mathcal{F})]$, and by Lemma 3 we have

$$
\begin{aligned}
\mathop{\mathbb{E}}_{S\sim\mathbb{P}^{|\mathcal{D}|}}[\mathcal{B}(\mathcal{F})] &\leq 2\mathop{\mathbb{E}}_{S\sim\mathbb{P}^{|\mathcal{D}|}}\mathbb{E}_{\boldsymbol{\sigma}\sim\{-1,+1\}^{|\mathcal{D}|}}\sup_{f_\theta\in B_K(M)}\left[\frac{1}{|\mathcal{D}|}\sum_{(u,i)\in\mathcal{D}}\sigma_{u,i}o_{u,i}\hat{w}_{u,i}e_{u,i}\right] \\
&= 2\mathop{\mathbb{E}}_{S\sim\mathbb{P}^{|\mathcal{D}|}}\{\mathcal{R}(\mathcal{L}_{\text{KBIPS}}(\theta_r))\},
\end{aligned}
$$

where

$$\mathcal{R}(\mathcal{L}_{\text{KBIPS}}(\theta_r)) := \mathbb{E}_{\boldsymbol{\sigma}\sim\{-1,+1\}^{|\mathcal{D}|}}\sup_{f_\theta\in B_K(M)}\left[\frac{1}{|\mathcal{D}|}\sum_{(u,i)\in\mathcal{D}}\sigma_{u,i}o_{u,i}\hat{w}_{u,i}e_{u,i}\right].$$

By applying McDiarmid's inequality in Lemma 4 and the assumptions that $\hat{w}_{u,i} \leq C$, also note that $\hat{r}_{u,i} \leq \sup_x \sqrt{K(x,x)}\|f\|_\mathcal{F} \leq MB$, thus $e_{u,i} = \mathcal{L}(\hat{r}_{u,i}, r_{u,i}) \leq E_0 + LMB$, let

$$c = \frac{2C(E_0 + LMB)}{|\mathcal{D}|},$$

then with probability at least $1 - \frac{\eta}{2}$,

$$
\begin{aligned}
\left|\mathcal{R}(\mathcal{L}_{\text{KBIPS}}(\theta_r)) - \mathop{\mathbb{E}}_{S\sim\mathbb{P}^{|\mathcal{D}|}}\{\mathcal{R}(\mathcal{L}_{\text{KBIPS}}(\theta_r))\}\right| &\leq \frac{2C(E_0 + LMB)}{|\mathcal{D}|}\sqrt{\frac{\log(4/\eta)|\mathcal{D}|}{2}} \\
&= 2C(E_0 + LMB)\sqrt{\frac{\log(4/\eta)}{2|\mathcal{D}|}}.
\end{aligned}
$$

By the assumption that $\hat{w}_{u,i} \leq C$ and $\delta(r, \cdot)$ is $L$-Lipschitz continuous for all $r$, we have

$$\mathcal{R}(\mathcal{L}_{\text{KBIPS}}(\theta_r)) \leq CL\mathcal{R}(\mathcal{F}) \leq \frac{CLMB}{\sqrt{|\mathcal{D}|}},$$

where the first inequality is from Lemma 5 and Lemma 6 with $L$ as Lipschitz constant, the second inequality is from Lemma 7, and $\mathcal{R}(\mathcal{F})$ is the empirical Rademacher complexity

$$
\mathcal{R}(\mathcal{F}) = \mathbb{E}_{\boldsymbol{\sigma} \sim \{-1,+1\}^{|\mathcal{D}|}} \sup_{f_\theta \in B_K(M)} \left[ \frac{1}{|\mathcal{D}|} \sum_{(u,i) \in \mathcal{D}} \sigma_{u,i} f(x_{u,i}) \right],
$$

where $\boldsymbol{\sigma} = \{\sigma_{u,i} : (u,i) \in \mathcal{D}\}$, and $\sigma_{u,i}$ are independent uniform random variables taking values in $\{-1,+1\}$. The random variables $\sigma_{u,i}$ are called Rademacher variables.

For the rest term $\mathcal{B}(\mathcal{F}) - \mathbb{E}_{S \sim \mathbb{P}^{|\mathcal{D}|}}[\mathcal{B}(\mathcal{F})]$, by applying McDiarmid's inequality in Lemma 4 and the assumptions that $\hat{w}_{u,i} \leq C$ and $e_{u,i} \leq E_0 + LMB$, let

$$
c = \frac{C(E_0 + LMB)}{|\mathcal{D}|},
$$

then with probability at least $1 - \frac{\eta}{2}$,

$$
\left| \mathcal{B}(\mathcal{F}) - \mathbb{E}_{S \sim \mathbb{P}^{|\mathcal{D}|}}[\mathcal{B}(\mathcal{F})] \right| \leq \frac{C(E_0 + LMB)}{|\mathcal{D}|} \sqrt{\frac{\log(4/\eta)|\mathcal{D}|}{2}} = C(E_0 + LMB)\sqrt{\frac{\log(4/\eta)}{2|\mathcal{D}|}}.
$$

We now bound $\mathcal{B}(\mathcal{F})$ by combining the above results. Formally, with probability at least $1 - \eta$,

$$
\begin{aligned}
\mathcal{B}(\mathcal{F}) &= \mathbb{E}_{S \sim \mathbb{P}^{|\mathcal{D}|}}[\mathcal{B}(\mathcal{F})] + \left\{ \mathcal{B}(\mathcal{F}) - \mathbb{E}_{S \sim \mathbb{P}^{|\mathcal{D}|}}[\mathcal{B}(\mathcal{F})] \right\} \\
&\leq 2 \mathbb{E}_{S \sim \mathbb{P}^{|\mathcal{D}|}} \{ \mathcal{R}(\mathcal{L}_{\text{KBIPS}}(\theta_r)) \} + \left\{ \mathcal{B}(\mathcal{F}) - \mathbb{E}_{S \sim \mathbb{P}^{|\mathcal{D}|}}[\mathcal{B}(\mathcal{F})] \right\} \\
&\leq 2 \mathcal{R}(\mathcal{L}_{\text{KBIPS}}(\theta_r)) + 4C(E_0 + LMB)\sqrt{\frac{\log(4/\eta)}{2|\mathcal{D}|}} + \left\{ \mathcal{B}(\mathcal{F}) - \mathbb{E}_{S \sim \mathbb{P}^{|\mathcal{D}|}}[\mathcal{B}(\mathcal{F})] \right\} \\
&\leq 2 \mathcal{R}(\mathcal{L}_{\text{KBIPS}}(\theta_r)) + 5C(E_0 + LMB)\sqrt{\frac{\log(4/\eta)}{2|\mathcal{D}|}} \\
&\leq \frac{2CLMB}{\sqrt{|\mathcal{D}|}} + 5C(E_0 + LMB)\sqrt{\frac{\log(4/\eta)}{2|\mathcal{D}|}}.
\end{aligned}
$$

We now bound the ideal loss by combining the above results. Formally, with probability at least $1 - \eta$, we have

$$
\begin{aligned}
\mathcal{L}_{\text{Ideal}}(\theta_r) &\leq \mathcal{L}_{\text{KBIPS}}(\theta_r) + |\text{Bias}(\mathcal{L}_{\text{KBIPS}}(\theta_r))| + \mathcal{B}(\mathcal{F}) \\
&\leq \mathcal{L}_{\text{KBIPS}}(\theta_r) + |\text{Bias}(\mathcal{L}_{\text{KBIPS}}(\theta_r))| + \frac{2CLMB}{\sqrt{|\mathcal{D}|}} + 5C(E_0 + LMB)\sqrt{\frac{\log(4/\eta)}{2|\mathcal{D}|}}.
\end{aligned}
$$

In Theorem 1, we have already proved that

$$
|\text{Bias}(\mathcal{L}_{\text{KBIPS}}(\theta_r))| = \frac{1}{|\mathcal{D}|} \left| \sum_{(u,i) \in \mathcal{D}} (o_{u,i} \hat{w}_{u,i} - 1) e_{u,i} \right|,
$$

therefore with probability at least $1 - \eta$, we have

$$
\mathcal{L}_{\text{Ideal}}(\theta_r) \leq \mathcal{L}_{\text{KBIPS}}(\theta_r) + |\text{Bias}(\mathcal{L}_{\text{KBIPS}}(\theta_r))| + \frac{2CLMB}{\sqrt{|\mathcal{D}|}} + 5C(E_0 + LMB)\sqrt{\frac{\log(4/\eta)}{2|\mathcal{D}|}}.
$$

We then prove the generalization bound of the KBDR estimator, similarly, the ideal loss can be decomposed as follows.

$$
\begin{aligned}
\mathcal{L}_{\text{Ideal}}(\theta_r) &= \mathcal{L}_{\text{KBDR}}(\theta_r) + (\mathcal{L}_{\text{Ideal}}(\theta_r) - \mathbb{E}(\mathcal{L}_{\text{KBDR}}(\theta_r))) + (\mathbb{E}(\mathcal{L}_{\text{KBDR}}(\theta_r)) - \mathcal{L}_{\text{KBDR}}(\theta_r)) \\
&= \mathcal{L}_{\text{KBDR}}(\theta_r) + \text{Bias}(\mathcal{L}_{\text{KBDR}}(\theta_r)) + (\mathbb{E}(\mathcal{L}_{\text{KBDR}}(\theta_r)) - \mathcal{L}_{\text{KBDR}}(\theta_r)) \\
&\leq \mathcal{L}_{\text{KBDR}}(\theta_r) + |\text{Bias}(\mathcal{L}_{\text{KBDR}}(\theta_r))| \\
&\quad + \sup_{f_\theta \in B_K(M)} \left( \mathbb{E}\left[ \frac{1}{|\mathcal{D}|} \sum_{(u,i)\in\mathcal{D}} \left[ \hat{e}_{u,i} + o_{u,i}\hat{w}_{u,i}(e_{u,i} - \hat{e}_{u,i}) \right] \right] - \frac{1}{|\mathcal{D}|} \sum_{(u,i)\in\mathcal{D}} \left[ \hat{e}_{u,i} + o_{u,i}\hat{w}_{u,i}(e_{u,i} - \hat{e}_{u,i}) \right] \right).
\end{aligned}
$$

For simplicity, we denote the last term in the above formula as

$$
\mathcal{B}(\mathcal{F}) = \sup_{f_\theta \in B_K(M)} \left( \mathbb{E}\left[ \frac{1}{|\mathcal{D}|} \sum_{(u,i)\in\mathcal{D}} \left[ \hat{e}_{u,i} + o_{u,i}\hat{w}_{u,i}(e_{u,i} - \hat{e}_{u,i}) \right] \right] - \frac{1}{|\mathcal{D}|} \sum_{(u,i)\in\mathcal{D}} \left[ \hat{e}_{u,i} + o_{u,i}\hat{w}_{u,i}(e_{u,i} - \hat{e}_{u,i}) \right] \right),
$$

we then aim to bound $\mathcal{B}(\mathcal{F})$ in the following.

Note that

$$
\mathcal{B}(\mathcal{F}) = \mathbb{E}_{S\sim\mathbb{P}^{|\mathcal{D}|}}[\mathcal{B}(\mathcal{F})] + \left\{ \mathcal{B}(\mathcal{F}) - \mathbb{E}_{S\sim\mathbb{P}^{|\mathcal{D}|}}[\mathcal{B}(\mathcal{F})] \right\},
$$

where the first term is $\mathbb{E}_{S\sim\mathbb{P}^{|\mathcal{D}|}}[\mathcal{B}(\mathcal{F})]$, and by Lemma 3 we have

$$
\begin{aligned}
\mathbb{E}_{S\sim\mathbb{P}^{|\mathcal{D}|}}[\mathcal{B}(\mathcal{F})] &\leq 2 \mathbb{E}_{S\sim\mathbb{P}^{|\mathcal{D}|}} \mathbb{E}_{\boldsymbol{\sigma}\sim\{-1,+1\}^{|\mathcal{D}|}} \sup_{f_\theta \in B_K(M)} \left[ \frac{1}{|\mathcal{D}|} \sum_{(u,i)\in\mathcal{D}} \sigma_{u,i}\left[ \hat{e}_{u,i} + o_{u,i}\hat{w}_{u,i}(e_{u,i} - \hat{e}_{u,i}) \right] \right] \\
&= 2 \mathbb{E}_{S\sim\mathbb{P}^{|\mathcal{D}|}} \{\mathcal{R}(\mathcal{L}_{\text{KBDR}}(\theta_r))\},
\end{aligned}
$$

where

$$
\mathcal{R}(\mathcal{L}_{\text{KBDR}}(\theta_r)) := \mathbb{E}_{\boldsymbol{\sigma}\sim\{-1,+1\}^{|\mathcal{D}|}} \sup_{f_\theta \in B_K(M)} \left[ \frac{1}{|\mathcal{D}|} \sum_{(u,i)\in\mathcal{D}} \sigma_{u,i}\left[ \hat{e}_{u,i} + o_{u,i}\hat{w}_{u,i}(e_{u,i} - \hat{e}_{u,i}) \right] \right].
$$

By applying McDiarmid's inequality in Lemma 4 and the assumptions that $\hat{w}_{u,i} \leq C$, $\hat{e}_{u,i} \leq E_0 + LMB$, and $e_{u,i} \leq E_0 + LMB$, let

$$
c = \frac{2(E_0 + LMB)(1 + 2C)}{|\mathcal{D}|},
$$

then with probability at least $1 - \frac{\eta}{2}$,

$$
\begin{aligned}
\left| \mathcal{R}(\mathcal{L}_{\text{KBDR}}(\theta_r)) - \mathbb{E}_{S\sim\mathbb{P}^{|\mathcal{D}|}} \{\mathcal{R}(\mathcal{L}_{\text{KBDR}}(\theta_r))\} \right| &\leq \frac{2(E_0 + LMB)(1 + 2C)}{|\mathcal{D}|} \sqrt{\frac{\log(4/\eta)|\mathcal{D}|}{2}} \\
&= 2(E_0 + LMB)(1 + 2C) \sqrt{\frac{\log(4/\eta)}{2|\mathcal{D}|}}.
\end{aligned}
$$

By the assumption that $\hat{w}_{u,i} \leq C$ and $\delta(r, \cdot)$ is $L$-Lipschitz continuous for all $r$, we have

$$
\mathcal{R}(\mathcal{L}_{\text{KBDR}}(\theta_r)) \leq L(1 + 2C)\mathcal{R}(\mathcal{F}) \leq (1 + 2C)\frac{LMB}{\sqrt{|\mathcal{D}|}},
$$

where the first inequality is from Lemma 5 and Lemma 6 with $L(1 + 2C)$ as Lipschitz constant, the second inequality is from Lemma 7, and $\mathcal{R}(\mathcal{F})$ is the empirical Rademacher complexity

$$
\mathcal{R}(\mathcal{F}) = \mathbb{E}_{\boldsymbol{\sigma}\sim\{-1,+1\}^{|\mathcal{D}|}} \sup_{f_\theta \in B_K(M)} \left[ \frac{1}{|\mathcal{D}|} \sum_{(u,i)\in\mathcal{D}} \sigma_{u,i} f(x_{u,i}) \right],
$$

where $\boldsymbol{\sigma} = \{\sigma_{u,i} : (u,i) \in \mathcal{D}\}$, and $\sigma_{u,i}$ are independent uniform random variables taking values in $\{-1, +1\}$. The random variables $\sigma_{u,i}$ are called Rademacher variables.

For the rest term $\mathcal{B}(\mathcal{F}) - \mathop{\mathbb{E}}\limits_{S \sim \mathbb{P}^{|\mathcal{D}|}} [\mathcal{B}(\mathcal{F})]$, by applying McDiarmid's inequality in Lemma 4 and the assumptions that $\hat{w}_{u,i} \leq C$, $\hat{e}_{u,i} \leq E_0 + LMB$, and $e_{u,i} \leq E_0 + LMB$, let

$$c = \frac{(E_0 + LMB)(1 + 2C)}{|\mathcal{D}|},$$

then with probability at least $1 - \frac{\eta}{2}$,

$$\left| \mathcal{B}(\mathcal{F}) - \mathop{\mathbb{E}}\limits_{S \sim \mathbb{P}^{|\mathcal{D}|}} [\mathcal{B}(\mathcal{F})] \right| \leq \frac{(E_0 + LMB)(1 + 2C)}{|\mathcal{D}|} \sqrt{\frac{\log(4/\eta)|\mathcal{D}|}{2}} = (E_0 + LMB)(1 + 2C) \sqrt{\frac{\log(4/\eta)}{2|\mathcal{D}|}}.$$

We now bound $\mathcal{B}(\mathcal{F})$ by combining the above results. Formally, with probability at least $1 - \eta$,

$$\mathcal{B}(\mathcal{F}) = \mathop{\mathbb{E}}\limits_{S \sim \mathbb{P}^{|\mathcal{D}|}} [\mathcal{B}(\mathcal{F})] + \left\{ \mathcal{B}(\mathcal{F}) - \mathop{\mathbb{E}}\limits_{S \sim \mathbb{P}^{|\mathcal{D}|}} [\mathcal{B}(\mathcal{F})] \right\}$$

$$\leq 2 \mathop{\mathbb{E}}\limits_{S \sim \mathbb{P}^{|\mathcal{D}|}} \{\mathcal{R}(\mathcal{L}_{\text{KBDR}}(\theta_r))\} + \left\{ \mathcal{B}(\mathcal{F}) - \mathop{\mathbb{E}}\limits_{S \sim \mathbb{P}^{|\mathcal{D}|}} [\mathcal{B}(\mathcal{F})] \right\}$$

$$\leq 2\mathcal{R}(\mathcal{L}_{\text{KBDR}}(\theta_r)) + 4(E_0 + LMB)(1 + 2C) \sqrt{\frac{\log(4/\eta)}{2|\mathcal{D}|}} + \left\{ \mathcal{B}(\mathcal{F}) - \mathop{\mathbb{E}}\limits_{S \sim \mathbb{P}^{|\mathcal{D}|}} [\mathcal{B}(\mathcal{F})] \right\}$$

$$\leq 2\mathcal{R}(\mathcal{L}_{\text{KBDR}}(\theta_r)) + 5(E_0 + LMB)(1 + 2C) \sqrt{\frac{\log(4/\eta)}{2|\mathcal{D}|}}$$

$$\leq 2(1 + 2C) \frac{LMB}{\sqrt{|\mathcal{D}|}} + 5(E_0 + LMB)(1 + 2C) \sqrt{\frac{\log(4/\eta)}{2|\mathcal{D}|}}$$

$$= (1 + 2C) \left( \frac{2LMB}{\sqrt{|\mathcal{D}|}} + 5(E_0 + LMB) \sqrt{\frac{\log(4/\eta)}{2|\mathcal{D}|}} \right).$$

We now bound the ideal loss by combining the above results. Formally, with probability at least $1 - \eta$, we have

$$\mathcal{L}_{\text{Ideal}}(\theta_r) \leq \mathcal{L}_{\text{KBDR}}(\theta_r) + |\text{Bias}(\mathcal{L}_{\text{KBDR}}(\theta_r))| + \mathcal{B}(\mathcal{F})$$

$$\leq \mathcal{L}_{\text{KBDR}}(\theta_r) + |\text{Bias}(\mathcal{L}_{\text{KBDR}}(\theta_r))| + (1 + 2C) \left( \frac{2LMB}{\sqrt{|\mathcal{D}|}} + 5(E_0 + LMB) \sqrt{\frac{\log(4/\eta)}{2|\mathcal{D}|}} \right).$$

In Theorem 1, we have already proved that

$$|\text{Bias}(\mathcal{L}_{\text{KBDR}}(\theta_r))| = \frac{1}{|\mathcal{D}|} \left| \sum_{(u,i) \in \mathcal{D}} (o_{u,i} \hat{w}_{u,i} - 1)(e_{u,i} - \hat{e}_{u,i}) \right|,$$

therefore with probability at least $1 - \eta$, we have

$$\mathcal{L}_{\text{Ideal}}(\theta_r) \leq \mathcal{L}_{\text{KBDR}}(\theta_r) + |\text{Bias}(\mathcal{L}_{\text{KBDR}}(\theta_r))| + (1 + 2C) \left( \frac{2LMB}{\sqrt{|\mathcal{D}|}} + 5(E_0 + LMB) \sqrt{\frac{\log(4/\eta)}{2|\mathcal{D}|}} \right),$$

which yields the stated results. $\qquad \square$

