# OpenReview forum: "Debiased Collaborative Filtering with Kernel-Based Causal Balancing"
_ICLR.cc/2024/Conference — ICLR 2024 spotlight_

### Official Review · Reviewer_ptof · 2023-10-29

**Soundness:** 4 excellent
**Presentation:** 3 good
**Contribution:** 3 good
**Rating:** 8
**Confidence:** 3

**Summary:**

The paper proposes a unified kernel-based method to balance functions on the reproducing kernel Hilbert space (RKHS) to address the bias in collaborative filtering due to the observational nature of the collected data. Although balancing (i.e., re-weighting the sample loss) is an important technique to address such selection bias, this paper characterizes the effect of balancing low-dimensional functions on the bias of inverse propensity score (IPS) and doubly robust (DR) methods. A novel adaptive causal balancing method that alternates between unbiased evaluation and model training is proposed. Extensive numerical experiments are conducted on real-world recommendation data sets to demonstrate the proposed method could effectively improve prediction performance compared with the existing benchmarks.

**Strengths:**

1. The problem studied is important and relevant.
2. The idea is interesting and novel.
3. The evaluations are solid and convincing.

**Weaknesses:**

1. The optimization formulation for balancing may be computationally intractable if the number of items and/or the number of users are large.

2. It is unclear how to incorporate the proposed method into the modern deep learning (DL) based recommender system and to test its effectiveness in a field setting.

**Questions:**

1. How to update the weights for the problems where the number of items and/or the number of users are large?

2. Modern DL-based recommender system typically generates an embedding for each user and each item for subsequent tasks such as score prediction and recommendation. Is it possible to adjust this method as a means of fine-tuning the item and/or user embeddings?

---

> ### Author Response · Authors · 2023-11-23
> **Please kindly find our concise and clear rebuttal below for addressing your current concerns [W1, W2, Q1, Q2]**
>
> We sincerely appreciate the reviewer’s great efforts and insightful comments to improve our manuscript. In below, we address these concerns point by point and try our best to update the manuscript accordingly.
>
> > **[W1 & Q1] How to update the weights for the problems where the number of items and/or the number of users are large?**
>
> **Response:** We thank the reviewer for raising an interesting concern.
>
> - First, **the balancing weights $w_{u, i}$ are parameterized using a deep neural network model $g(x_{u, i}; \phi_w)$**, which ensures the model scalability and the model parameters $\phi_w$ could be much less than the number of user-item pairs when the number of items and/or the number of users is large.
>
> - Second, **we can write the primal optimization problem to the Lagrange dual problem**, in which the number of parameters is the same as the number of constraints $O(J)$, and is independent with the user numbers and item numbers.
>
> - Third, for the optimization problem, **we can uniformly sample a batch in $\mathcal{D} = \[u_1, u_2, ..., u_m\] \times \[i_1, i_2, ..., i_n\]$ to compute $\tau^{(j)}$ for $j = 1, 2, ..., J$**, instead of using all $m \cdot n$ user-item pairs that concerns the computational efficiency.
>
> > **[W2] How to incorporate the proposed method into the modern deep learning (DL) based recommender system?**
>
> **Response:** We thank the reviewer for the useful comments. **The proposed methods are model-agnostic**, which means the balancing weight model $g(x_{u, i}; \phi_w)$ and the prediction model $f(x_{u, i}; \theta)$ can adopt the modern deep neural networks. To show this, **we add the experiment on a large-scale industrial dataset Product** using **neural collaborative filtering (NCF) [1]** as the base model to empirically show the performance of the proposed methods in Appendix B, Table 2. The results are shown below.
> ||AUC|NDCG@20|F1@20|
> |:--:|:--:|:--:|:--:|
> |NCF| 0.823±0.001|0.575±0.002|0.166±0.002|
> |+CVIB|0.820±0.002|0.544±0.004|0.162±0.002|
> ||
> |+IPS|0.822±0.003|0.579±0.006|0.169±0.003|
> |+SNIPS|0.833±0.002|0.586±0.002|0.178±0.002|
> |+ASIPS|0.832±0.002|0.583±0.003|0.178±0.002|
> |+IPS-V2|0.835±0.001|0.588±0.002|0.181±0.001|
> |+CBIPS|0.832±0.001|0.584±0.003|0.178±0.003|
> |+WKBIPS|0.833±0.002| 0.588±0.003|0.181±0.003|
> |+AKBIPS|**0.836*±0.001**| **0.592*±0.002**|**0.183*±0.002**|
> ||
> |+DR |0.758±0.004|0.526±0.003|0.146±0.003|
> |+DR-JL|0.832±0.001|0.581±0.003|0.178±0.002|
> |+MRDR-JL|0.833±0.001|0.585±0.002|0.179±0.001|
> |+DR-BIAS|0.834±0.002|0.585±0.003|0.178±0.002|
> |+DR-MSE|0.834±0.002|0.587±0.002|0.180±0.001|
> |+MR|0.837±0.001|0.588±0.002|0.181±0.002|
> |+TDR-JL|0.834±0.002|0.582±0.002|0.179±0.003|
> |+SDR|0.835±0.002|0.587±0.003|0.179±0.002|
> |+DR-V2|0.837±0.002|0.586±0.004|0.182±0.002|
> |+CBDR|0.834±0.002 |0.586±0.003|0.179±0.003|
> |+WCBDR|0.837±0.002| 0.588±0.003|0.180±0.002|
> |+ACBDR|**0.840*±0.002**|**0.590*±0.003**|**0.183±0.002**|
> ||
>
> The experiment results show that the proposed kernel balancing methods achieve overall performance improvement compared to the baseline methods.
>
> > **[Q2] Is it possible to adjust this method as a means of fine-tuning the item and/or user embeddings?**
>
> **Response:** **Yes, it is possible to fine-tune the embeddings using our proposed methods**. We illustrate this by using **click-through rate (CTR) prediction** and **post-click conversion rate (pCVR) prediction** in e-commerce recommendation as an example. From a debiasing formulation, the data formats and the quantity of interest in the "offline rating prediction" scenario and the "e-commerce conversion rate prediction" scenario are exactly the same [2-4]. We provide a comparison table as below.
>
> |Terminology|Offline ratings prediction|E-commerce pCVR prediction|Comments|
> |:--:|:--:|:--:|:--:|
> |$x_{u, i}$|user and item features|user and item features|all can be obtained in the raw data|
> |$o_{u, i}$|rating observe indicator	|click indicator|$o_{u, i} = 1$ if observing, else $o_{u, i}=0$ if missing|
> |$r_{u, i}$|true rating, i.e., the true preference of user to item| post-click conversion indicator|only can be observed when $o_{u, i}=1$, otherwise missing|
> |$\hat{p}_{u, i}$|propensity model|CTR model|for both we have $p_{u, i} = P(o_{u, i} = 1 \mid x_{u, i})$|
> ||
>
> Therefore, we can adopt the method proposed in this paper to adaptively choose $\phi(x_{u, i})$ as a kernel function $K(x_{s, t}, x_{u, i})$ with some fixed $(s, t)$ and then fine-tune the CTR model $\hat{p}(o = 1|x_{u, i})$. **One possible way to fine-tune the CTR model is minimizing the directed Kullback entropy divergence [5] defined by $h(w_i) = w_i log({w_i}/{q_i})$ between the new and the original CTR model predictions** as the **objective function** in the proposed entropy-based optimization problem. Moreover, we can further fine-tune the pCVR model based on the fine-tuned CTR model.

---

> > ### Author Response · Authors · 2023-11-23
> > **Please kindly find our concise and clear rebuttal below for addressing your current concerns [References]**
> >
> > **We hope the above discussion will fully address your concerns about our work.** We look forward to your insightful and constructive responses to further help us improve the quality of our work. Thank you!
> >
> > ***
> > **References**
> >
> > [1] Xiangnan He et al. Neural collaborative filtering. WWW 2017.
> >
> > [2] Quanyu Dai et al. A generalized doubly robust learning framework for debiasing post-click conversion rate prediction. KDD 2022.
> >
> > [3] Siyuan Guo et al. Enhanced Doubly Robust Learning for Debiasing Post-Click Conversion Rate Estimation. SIGIR 21.
> >
> > [4] Hao Wang et al. ESCM^{2}: Entire Space Counterfactual Multi-Task Model for Post-Click Conversion Rate Estimation. SIGIR 22.
> >
> > [5] Kullback, S. Information theory and statistics. New York: Wiley 1959.

---

### Official Review · Reviewer_4dU2 · 2023-10-29

**Soundness:** 3 good
**Presentation:** 2 fair
**Contribution:** 3 good
**Rating:** 6
**Confidence:** 3

**Summary:**

This work contributes to the field of debiased recommendation recommendation by proposing a
universal kernel-based balancing method to balance functions. It also provides theoretical analysis and guidance of balancing property under finite samples. Experimental results on three real-world datasets verify the proposed balancing methods.

**Strengths:**

1.This work discusses the limitations of existing methods including the Inverse Propensity Score (IPS) and Doubly Robust (DR) method for debiased recommendation, showing that the importance of balancing property under finite-dimensional function classes.

2.The authors extend them to CBIPS and CBDR estimators and it is reasonable to proposed kernel balancing method for optimization.

3.The work also provides theoretical analysis and proof of the proposed kernel balancing and proposes three causal balancing methods to effectively balance the kernel functions.

4.Experimental results show the effectiveness of the adaptive kernel balancing method.

**Weaknesses:**

1.It is advisable to include a discussion of the relationships or distinctions between this work and prior research in the Related Work section.

2.In terms of the Adaptive Kernel Balancing method, I am confused why balancing the kernel functions with maximal $|\alpha_{s,t}|$ can contribute the most to the $e_{u,i}$. It requires more explanation to improve the readability. Besides, the authors adopt this way to improve the efficiency but there is no corresponding experiment to validate this.

3.Writting of this paper needs improving:
- The background of debias recommendation is not enough.
- The definition of $o_{u,i}$ is not clarified clearly.
- Grammar needs carefully checking:
  - the usage of “an” such as “Moreover, we propose an causal”
  - learn an appropriate propensity model that achieve lower estimation bias.

**Questions:**

See above.

---

> ### Author Response · Authors · 2023-11-23
> **Please kindly find our concise and clear rebuttal below for addressing your current concerns**
>
> We sincerely appreciate the reviewer’s great efforts and insightful comments to improve our manuscript. In below, we address these concerns point by point and try our best to update the manuscript accordingly.
>
> > **[W1] It is advisable to include a discussion of the relationships or distinctions between this work and prior research in the Related Work section.**
>
> **Response:** Thank you for your kind advice. As suggested by the reviewer, **we add a discussion of the relationships and distinctions between our work and prior research** in the Related Work section **for both “Debiased Recommendation” and “Covariate Balancing in Causal Inference” parts.**
>
> > **[W2.1] Why balancing the kernel functions with maximal $|\alpha_{s, t}|$ can contribute the most to the $|e_{u, i}|$?**
>
> **Response:** Taking CBIPS as an example, in the last equation on Page 5, we use the universal property and the representer theorem to show the bias of CBIPS estimator is
>
> $$\operatorname{Bias}^2(\mathcal{L}_{\mathrm{CBIPS}}(\theta))$$
>
>  $$= \[\frac{1}{|\mathcal{D}|}\sum_{(u,i) \in \mathcal{D}} (o_{u, i} w_{u,i}-1)(\sum_{(s, t) \in \mathcal{D}}\alpha_{s, t}K(x, x_{s, t})) \]^2$$
>  $$= \[\frac{1}{|\mathcal{D}|}\sum_{(s, t) \in \mathcal{D}}\alpha_{s, t}\sum_{(u,i) \in \mathcal{D}} (o_{u, i} w_{u,i}-1)(K(x, x_{s, t})) \]^2$$
>  $$= \[\frac{1}{|\mathcal{D}|}\sum_{(s, t) \in \mathcal{D}}\alpha_{s, t}\sum_{(u,i) \in \mathcal{D}} (o_{u, i} w_{u,i}K(x, x_{s, t}) - K(x, x_{s, t})) \]^2.$$
> If we balance $K(x, x_{s, t})$, we have $$\frac{1}{|\mathcal{D}|}\sum_{  (u,i) \in \mathcal{D} }  o_{u, i}w_{u,i} K(x, x_{s, t})  =   \frac{1}{|\mathcal{D}|}\sum_{ (u,i) \in \mathcal{D} }  K(x, x_{s, t}),$$ that is,
> $$\frac{1}{|\mathcal{D}|}\sum_{(u,i) \in \mathcal{D}} (o_{u, i} w_{u,i}K(x, x_{s, t}) - K(x, x_{s, t})) = 0.$$
>
> Therefore, selecting the $K(x, x_{s, t})$ with maximal $|\alpha_{s, t}|$ to balance can effectively and rapidly reduce $\operatorname{Bias}(\mathcal{L}_{\mathrm{CBIPS}}(\theta))$.
>
> > **[W2.2] No corresponding experiment to validate the efficiency?**
>
> **Response:** Thanks for pointing out this issue and we really apologize for the **typo** here. **We intend to say "effectiveness" instead of "efficiency".** Nevertheless, we still **add the experiments to analyze the trade-off between the effectiveness and the efficiency of CBDR and ACBDR** in Appendix B, Table 3. The results are shown below.
>
> | Number of balancing functions | Metrics | CBDR-Gaussian | ACBDR-Gaussian |
> |:--:|:--:|:--:|:--:|
> |$J=3$| AUC     | 0.681±0.003   | 0.688±0.003    |
> || NDCG@5  | 0.650±0.003   | 0.657±0.003    |
> || F1@5    | 0.323±0.002   | 0.326±0.002    |
> || Time(s) | 389.28±17.51  | 664.99±12.37   |
> ||
> |$J=5$| AUC     | 0.683±0.002   | 0.694±0.002    |
> || NDCG@5  | 0.652±0.003   | 0.664±0.002    |
> || F1@5    | 0.325±0.002   | 0.332±0.002    |
> || Time(s) | 394.51±16.61  | 678.13±15.43   |
> ||
> |$J=10$| AUC     | 0.682±0.002   | 0.694±0.002    |
> || NDCG@5  | 0.650±0.003   | 0.663±0.002    |
> || F1@5    | 0.324±0.003   | 0.330±0.003    |
> || Time(s) | 399.91±8.58   | 719.63±23.79   |
> ||
> |$J=20$                           | AUC     | 0.683±0.002   | 0.695±0.002    |
> || NDCG@5  | 0.651±0.002   | 0.664±0.003    |
> || F1@5    | 0.325±0.002   | 0.332±0.003    |
> || Time(s) | 389.11±22.39  | 727.29±26.76   |
> ||
> |$J=50$ | AUC     | 0.684±0.002   | 0.695±0.002    |
> || NDCG@5  | 0.652±0.003   | 0.664±0.002    |
> || F1@5    | 0.324±0.003   | 0.333±0.002    |
> || Time(s) | 407.89±11.67  | 722.43±25.55   |
> ||
>
> The ACBDR method stably outperforms CBDR with varying number of balancing functions, which shows the **effectiveness** of the proposed adaptively balancing method. **For efficiency, ACBDR takes about "doubled running time" to converge than KBDR,** since it needs to sort all the coefficients of kernel functions, which is time-consuming when the number of user-item pairs is large. When $J > 5$, the performance of the AKBDR does not improve as the growth of $J$, but the running time increases.
>
> > **[W3] Writting of this paper needs improving.**
>
> **Response:** We thank the reviewer for pointing out this issue, and **we have added the formal definition of $o_{u, i}$ in Section 3.1, and carefully polished the manuscript in our revised version to avoid typos and improve readability.**
>
> ***
> **We hope the above discussion will fully address your concerns about our work, and we would really appreciate it if you could be generous in raising your score.** We look forward to your insightful and constructive responses to further help us improve the quality of our work. Thank you!

---

### Official Review · Reviewer_CJCu · 2023-10-30

**Soundness:** 3 good
**Presentation:** 3 good
**Contribution:** 3 good
**Rating:** 6
**Confidence:** 4

**Summary:**

This study explores the concept of incorporating a balance regularization loss to mitigate selection bias in Recommender Systems (RS). It centers on determining which function class requires balancing. The authors reframe the issue as an optimization problem, contending that the prediction error should result from a linear combination of balancing functions, supported by theoretical analysis. To achieve this, the authors suggest utilizing kernel functions as balancing functions and put forth three approaches for selecting these kernels. Empirical experiments illustrate the benefits of the proposed balancing techniques.

**Strengths:**

- The study delves deeper into the challenge of addressing selection bias by examining the specific function that warrants balance. This research problem introduces a novel perspective.

- The concept of employing a linear combination of kernel functions to attain both unbiasedness and universality is intriguing and offers inspiration.

**Weaknesses:**

I think the idea in this paper is interesting and has some value. However, I have some significant concerns that prevent me from recommending this manuscript for acceptance.

 **1. Motivation**

- W1: This paper lacks a sufficient motivation for the importance of carefully selecting the function class for balancing. The authors do discuss the computational cost associated with choosing numerous balancing functions in Section 3.2. However, the original paper [1] suggests that even selecting a single function for balancing can yield performance improvements. It is essential to elucidate why the choice of multiple balancing functions is necessary and why the specific type of function is critical. In my view, the motivation may be derived from Corollary 1 to some extent, which asserts that certain types of balancing functions introduce bias (correct me if I am wrong). Nonetheless, it would be valuable to provide a more insightful explanation in the introduction section. I would like to see an illustrative example, which could enhance comprehension of the motivation.

**2. Clarity**

This paper contains several areas of ambiguity and insufficient support for its claims, which hinder the comprehension of its ideas. Some key issues include:

- W2: The optimization problems in Sections 4.2 and 4.4 are perplexing. The rationale behind minimizing the sum of $g(\cdot) \log g(\cdot)$ as opposed to a standard cross-entropy loss is unclear and confusing. The purpose of constraining the sum of $o_{u,i}g(\cdot)$ to be one and the meaning of the third constraint require more detailed explanation and motivation. The authors should clarify the significance and reasoning behind each statement in the proposed optimization problem.

- W3: Several theorems and corollaries, such as Theorem 1, Corollary 1, and Lemma 2, lack supporting proofs. In particular, the proof of Corollary 1, which appears directly relevant to the paper's motivation, is crucial.

- W4: The paper suggests that "balancing propensity can reduce the generalization bound", but this claim is not clearly evident from Theorem 2, which seems to provide a bound without a comparative analysis.

- W5: The fourth paragraph in Section 1 references "the first question" and "the second question", but it is unclear where these questions are presented in the paper.

- W6: The paper does not elaborate on how the parameters within the kernel functions are learned. For instance, Gaussian kernel functions are known to have a parameter, $\sigma$. The paper should explain the process for selecting suitable parameter values.

- W7: Given that kernel functions are not frequently used for debiasing Recommender Systems, it is recommended that the authors provide more specific definitions and formulations regarding the kernel functions in Section 4.3 to enhance clarity.

- W8: There are some typos in this paper.
  - Section 1: "it is the first paper provides ...", "Ours theoretical analysis shows...".
  - Section 4.4: "which Random chooses...".

**3. Experiments**

- W9: The paper does not specify which kernel function was chosen, whether Gaussian or exponential.

- W10: I found that the performance on "Coat" you reported in Table 1 is significantly lower than that on [1]. It would be helpful to understand whether this variance is attributed to different experimental settings or other factors.

- W11: The abbreviation "CB" for causal balancing is indeed similar to "Covariate Balancing" and may cause confusion. A more distinct abbreviation should be considered to prevent any potential misunderstandings.

Overall, the motivation and clarity are my main concerns. I will consider changing my score if I receive a high-quality (concise and clear) rebuttal.

[1] Haoxuan Li, Yanghao Xiao, Chunyuan Zheng, Peng Wu, and Peng Cui. Propensity matters: Measuring and enhancing balancing for recommendation. 2023d.


---

*After the rebuttal: the authors address the most of my concerns. Therefore, I am glad to raise my score from 3 to 6.*

**Questions:**

Please see the Weaknesses part.

---

> ### Author Response · Authors · 2023-11-22
> **Please kindly find our concise and clear rebuttal below for addressing your current concerns [W1-W4]**
>
> We sincerely appreciate the reviewer’s great efforts and insightful comments to improve our manuscript. In below, we address these concerns point by point and try our best to update the manuscript accordingly.
>
> **1. Motivation**
>
> > **[W1] This paper lacks a sufficient motivation for the importance of carefully selecting the function class for balancing.** The authors do discuss the computational cost associated with choosing numerous balancing functions in Section 3.2. However, the original paper [1] suggests that even selecting a single function for balancing can yield performance improvements. It is essential to elucidate why the choice of multiple balancing functions is necessary and why the specific type of function is critical. In my view, the motivation may be derived from Corollary 1 to some extent, which asserts that certain types of balancing functions introduce bias (correct me if I am wrong). Nonetheless, it would be valuable to provide a more insightful explanation in the introduction section.
>
> **Response:** We thank the reviewer for the careful and insightful thoughts, and **have revised the Introduction Section** in our revised manuscript.
>
> -	First, we **agree** that “the original paper [1] suggests that even selecting a single function for balancing can yield performance improvements”, and **our experimental results also prove this.**
>
> -	However, on the one hand, **it is not realistic to balance all possible functions for a specific model** using only finite samples.
>
> -	On the other hand, **balancing only a single function is not sufficient for IPS and DR methods to achieve unbiased learning** (see Corollary 1 for the formal theoretical results).
>
> -	Thus, **it is necessary to discuss which functions should be more favored to be balanced for the IPS and DR estimators**, resulting in **smaller estimation biases of the ideal loss** and enhanced performance of unbiased learning.
>
> **2. Clarity**
>
> > **[W2] The optimization problems in Sections 4.2 and 4.4 are perplexing. The rationale behind minimizing the sum of $g(\cdot)\log g(\cdot)$ as opposed to a standard cross-entropy loss is unclear and confusing. The purpose of constraining the sum of $o_{u, i}g(\cdot)$ to be one and the meaning of the third constraint require more detailed explanation and motivation. The authors should clarify the significance and reasoning behind each statement in the proposed optimization problem.**
>
> **Response:** We thank the reviewer for pointing out this issue, and **have added detailed explanations regarding the optimization problems in Sections 4.2 and 4.4.** Both of the optimization problems consist of the following three features.
>
> -	First, **the objective function** is the **empirical entropy of the balancing weights**, by the principle of maximum entropy, it **reaches the maximum value when the balancing weights are uniform**, thus **effectively avoiding high variance due to extremely small propensities.**
>
> -	Second, **the balancing constraints** are imposed to equalize the selected covariate functions between the observed and missing samples.
>
> -	Third, **two normalization constraints are imposed**, which implies that the weights sum to the normalization constant of one, and the **nonnegativity of the balancing weights**, making the empirical entropy as the objective function well-defined.
>
> -	For the optimization problem in Section 4.4, we **further introduce a slack variable $\xi_j$** for each balancing function and **a pre-specified threshold $C$**, since **achieving strict balancing on all balancing functions is usually infeasible** as the number of balancing functions increases.
>
> > **[W3] Several theorems and corollaries, such as Theorem 1, Corollary 1, and Lemma 2, lack supporting proofs. In particular, the proof of Corollary 1, which appears directly relevant to the paper's motivation, is crucial.**
>
> **Response:** We really apologize for the lack of supporting proofs, and **have added detailed proofs for Theorem 1, Corollary 1, and Lemma 2 in Appendix A.**
>
> > **[W4] The paper suggests that "balancing propensity can reduce the generalization bound", but this claim is not clearly evident from Theorem 2, which seems to provide a bound without a comparative analysis.**
>
> **Response:** We thank the reviewer for pointing out this issue, and **have added detailed discussions in Section 4.5,** explaining the reason why the derived generalization bound in RKHS is able to be greatly reduced when adopting the proposed adaptive KBDR learning approach as in Alg. 1.
>
> -	For the **first term in the generalization bound**, the prediction model minimizes the loss $\mathcal{L}_{KBDR}(\theta)$ during the model training phase.
>
> -	For the **second term in the generalization bound**, as shown in Theorem 1 and Corollary 1, the proposed adaptive kernel balancing method can automatically choose the balancing functions that most need to be balanced to reduce the bias of the KBDR estimator.

---

> > ### Author Response · Authors · 2023-11-22
> > **Please kindly find our concise and clear rebuttal below for addressing your current concerns [W5-W8]**
> >
> > > **[W5] The fourth paragraph in Section 1 references "the first question" and "the second question", but it is unclear where these questions are presented in the paper.**
> >
> > **Response:** The reviewer has raised an important point, and **we have carefully revised the presentation here in Section 1** to improve the clarity.
> >
> > -	First, we propose universal kernel-based balancing methods to balance functions on the reproducing kernel Hilbert space (RKHS), which **adaptively selects the functions that most need to be balanced to reduce the estimation biases of the IPS and DR estimators**.
> >
> > -	Moreover, we propose **a novel entropy-based optimization problem to effectively balance the selected functions.**
> >
> > -	Finally, we perform **theoretical analysis** showing that the learned kernel balanced propensities can effectively **reduce the generalization bound**.
> >
> > > **[W6] The paper does not elaborate on how the parameters within the kernel functions are learned. For instance, Gaussian kernel functions are known to have a parameter, $\sigma$. The paper should explain the process for selecting suitable parameter values.**
> >
> > **Response:** To better validate the proposed methods with different kernels, we **add experiments using both Gaussian kernel and exponential kernel** to implement the proposed methods (in Table 1), as well as **clarify the missing kernel hyper-parameters tuning range** (in Section 5).
> >
> > -	The Gaussian kernel has the explicit form $$K^{Gau}\left(x, x^{\prime}\right)=\exp \left(-\frac{\left\|x-x^{\prime}\right\|^2}{2 \sigma^2}\right).$$
> >
> > -	The exponential kernel has the explicit form $$K^{Exp}\left(x, x^{\prime}\right)=\exp \left(-\frac{\left\|x-x^{\prime}\right\|}{2 \sigma^2}\right).$$
> >
> > -	**We tune the kernel hyper-parameter $\sigma^2$ in $\\{0.5, 1, 5\\}$ for both Gaussian and exponential kernels.**
> >
> > > **[W7] Given that kernel functions are not frequently used for debiasing Recommender Systems, it is recommended that the authors provide more specific definitions and formulations regarding the kernel functions in Section 4.3 to enhance clarity.**
> >
> > **Response:** Thank you for your kind advice. As suggested by the reviewer, we **add a formal definition of kernel function** at the beginning of Section 4.3, as well as **the explicit form of Gaussian kernel and exponential kernel as examples.**
> >
> > > **[W8] There are some typos in this paper.**
> >
> > **Response:** We thank the reviewer for pointing out this issue, and **we have carefully polished the whole manuscript in our revised version to avoid typos and improve readability.**

---

> > > ### Author Response · Authors · 2023-11-22
> > > **Please kindly find our concise and clear rebuttal below for addressing your current concerns [W9-W11]**
> > >
> > > **3. Experiments**
> > >
> > > > **[W9] Which kernel function was chosen?**
> > >
> > > **Response:** **The Gaussian kernel is chosen in our original manuscript.** In our revised version (on Page 8, Table 1), we report prediction performance results **using both Gaussian kernel and exponential kernel** on **all three datasets**, as shown below.
> > >
> > > |Coat Dataset|AUC|NDCG@5|F1@5|
> > > |:--:|:--:|:--:|:--:|
> > > |CBIPS (Exponential)|0.714±0.003 | 0.618±0.010 | 0.474±0.007 |
> > > |CBIPS (Gaussian) |0.715±0.005 | 0.619±0.010 | 0.475±0.008 |
> > > ||
> > > |  WCBIPS (Exponential)| 0.723±0.004 | 0.624±0.009 | 0.480±0.007 |
> > > |  WCBIPS (Gaussian) | 0.722±0.004 | 0.625±0.008 | 0.479±0.007 |
> > > ||
> > > |  ACBIPS (Exponential) | 0.732±0.004 | 0.636±0.006 | 0.483±0.006 |
> > > |  ACBIPS (Gaussian) | 0.730±0.003 | 0.633±0.008 | 0.484±0.007 |
> > > ||
> > > |CBDR (Exponential)  | 0.734±0.003 | 0.631±0.005 | 0.482±0.006 |
> > > |CBDR (Gaussian)  | 0.726±0.005 | 0.630±0.008 | 0.480±0.008 |
> > > ||
> > > |WCBDR (Exponential)  | 0.735±0.005 | 0.637±0.009 | 0.483±0.006 |
> > > | WCBDR (Gaussian)  | 0.732±0.003 | 0.638±0.007 | 0.483±0.005 |
> > > ||
> > > | ACBDR (Exponential)  | 0.745±0.004 | 0.645±0.008 | 0.493±0.007 |
> > > | ACBDR (Gaussian)  | 0.746±0.004 | 0.646±0.008 | 0.492±0.007 |
> > > ||
> > >
> > > |Music Dataset|AUC|NDCG@5|F1@5|
> > > |:--:|:--:|:--:|:--:|
> > > |CBIPS (Exponential)  | 0.676±0.002 | 0.642±0.003 | 0.318±0.002 |
> > > |CBIPS (Gaussian)  | 0.678±0.001 | 0.640±0.004 | 0.315±0.003 |
> > > ||
> > > |  WCBIPS (Exponential)  | 0.687±0.002 | 0.654±0.002 | 0.322±0.002 |
> > > |  WCBIPS (Gaussian)  | 0.686±0.002 | 0.650±0.002 | 0.321±0.002 |
> > > ||
> > > |ACBIPS (Exponential)  | 0.691±0.001 | 0.658±0.002 | 0.324±0.002 |
> > > |ACBIPS (Gaussian)  | 0.688±0.003 | 0.655±0.003 | 0.324±0.002 |
> > > ||
> > > |CBDR (Exponential)| 0.682±0.002 | 0.648±0.003 | 0.323±0.002 |
> > > |CBDR (Gaussian)| 0.683±0.002 | 0.652±0.003 | 0.325±0.002 |
> > > ||
> > > |WCBDR (Exponential)  | 0.685±0.003 | 0.654±0.003 | 0.325±0.002 |
> > > |WCBDR (Gaussian)  | 0.687±0.001 | 0.655±0.002 | 0.327±0.002 |
> > > ||
> > > |ACBDR (Exponential) |0.692±0.002 | 0.661±0.002 | 0.328±0.002 |
> > > |ACBDR (Gaussian)|0.694±0.002|0.664±0.002 | 0.332±0.002 |
> > > ||
> > >
> > > |Product Dataset|AUC|NDCG@20|F1@20|
> > > |:--:|:--:|:--:|:--:|
> > > |CBIPS (Exponential)|  0.763±0.001 | 0.463±0.007 | 0.134±0.002 |
> > > |CBIPS (Gaussian)|  0.760±0.003 | 0.470±0.008 | 0.133±0.003 |
> > > ||
> > > |WCBIPS (Exponential)|  0.765±0.003 | 0.475±0.007 | 0.136±0.003 |
> > > |WCBIPS (Gaussian)|  0.763±0.003 | 0.476±0.007 | 0.137±0.003 |
> > > ||
> > > |ACBIPS (Exponential)|  0.766±0.003 | 0.478±0.009 | 0.138±0.003 |
> > > |ACBIPS (Gaussian) |  0.767±0.003 | 0.480±0.009 | 0.139±0.003 |
> > > ||
> > > |CBDR (Exponential) | 0.765±0.004 | 0.460±0.006 | 0.138±0.003 |
> > > |    CBDR (Gaussian)|  0.766±0.003 | 0.469±0.007 | 0.134±0.004 |
> > > ||
> > > |    WCBDR (Exponential)|  0.773±0.003 | 0.489±0.008 | 0.142±0.003 |
> > > |    WCBDR (Gaussian) | 0.773 ±0.002 | 0.490±0.005 | 0.142±0.004 |
> > > ||
> > > |    ACBDR (Exponential) |  0.782±0.003 | 0.498±0.008 | 0.147±0.003 |
> > > |    ACBDR (Gaussian) |  0.782±0.005 | 0.503±0.006 | 0.148±0.004 |
> > > ||
> > >
> > > The results show that the proposed methods stably outperform the baseline methods in all metrics. Our methods perform similarly when adopting either the Gaussian kernel or the exponential kernel.
> > >
> > > > **[W10] Lower experimental results on "Coat" dataset.**
> > >
> > > **Response:** We thank the reviewer for pointing out this issue. We suppose the main reason is that **there exist several different experimental settings and code-base** in the field of debiased recommendation. Specifically, **we find the results in [1] are similar to the results in [2-4],** which use the following code-base: https://github.com/DongHande/AutoDebias. Instead, **we follow the experiment setting and code-base used in [5-7]**: https://github.com/RyanWangZf/CVIB-Rec, which leads to **similar results** compared to the published papers [5-7].
> > >
> > > > **[W11] Potential misunderstandings of the abbreviation "CB".**
> > >
> > > **Response:** Thanks for the comment and we fully agree with the reviewer. In our revised version, **we replaced causal balancing (CB) with kernel balancing (KB) and revised all figures and tables**, we think "KB" is a more specific and relevant abbreviation.
> > >
> > > ***
> > >
> > > **We hope the above discussion will fully address your concerns about our work, and we would really appreciate it if you could be generous in raising your score.** Thank you!
> > >
> > > ***
> > >
> > > > **References**
> > >
> > > [1] Haoxuan Li et al. Propensity matters: Measuring and enhancing balancing for recommendation. ICML 23.
> > >
> > > [2] Sihao Ding et al. Addressing Unmeasured Confounder for Recommendation with Sensitivity Analysis. KDD 22.
> > >
> > > [3] Jiawei Chen et al. AutoDebias: Learning to Debias for Recommendation. SIGIR 21.
> > >
> > > [4] Sihao Ding et al. Interpolative Distillation for Unifying Biased and Debiased Recommendation. SIGIR 22.
> > >
> > > [5] Zifeng Wang et al. Information Theoretic Counterfactual Learning from Missing-Not-At-Random Feedback. NeurIPS 22.
> > >
> > > [6] Haoxuan Li et al. TDR-CL: Targeted doubly robust collaborative learning for debiased recommendations. ICLR 23.
> > >
> > > [7] Haoxuan Li et al. StableDR: Stabilized doubly robust learning for recommendation on data missing not at random. ICLR 23.

---

> > > > ### Comment · Reviewer_CJCu · 2023-11-23
> > > > **Thank you for your responses**
> > > >
> > > > Dear authors,
> > > >
> > > > Thanks for the responses. The responses address the most of my concerns. While I still hope for an intuitive example or explanation illustrating why "balancing only a single function is not sufficient for IPS and DR methods to achieve unbiased learning", referencing Corollary 1 is enough for conveying the paper's motivation to readers.
> > > >
> > > > Additionally, please remember to update the paper's title to reflect the shift from causal balancing (CB) to kernel balancing (KB).
> > > >
> > > > Overall, I decide to raise my score to 6.

---

> ### Author Response · Authors · 2023-11-23
> **Thank you for the timely response and raising the score, below we offer an intuitive example**
>
> > **[Intuitive Example] While I still hope for an intuitive example or explanation illustrating why "balancing only a single function is not sufficient for IPS and DR methods to achieve unbiased learning", referencing Corollary 1 is enough for conveying the paper's motivation to readers.**
>
> **Response:** Let $\phi(X)$ be a vector of functions of $X$ to be balanced, such as $\phi(X)=\left(X, X^2\right)$. It assign a balancing weight $w_{u, i}$ to each observed user-item pair $\\{(u, i): o_{u, i}=1\\}$ by
> $$
> \begin{aligned}
> & \min_{w_{u, i}: o_{u, i}=1} \sum_{(u, i): o_{u, i}=1} w_{u, i} \log w_{u, i} \\
> & \text { subject to } w_{u, i} \geq 0 \\
> & \sum_{(u, i): o_{u, i}=1} w_{u, i}=1, \sum_{(u, i): o_{u, i}=1} w_{u, i} \phi\left(X_{u, i}\right)=\tilde{\phi}, \quad \text { (the balancing constraints)},
> \end{aligned}
> $$
> where $\tilde{\phi}=\frac{1}{|\mathcal{D}|} \sum_{(u, i)\in\mathcal{D}} \phi\left(X_{u, i}\right)$.
>
> - **Intuitively, if $e_{u, i}$ is a linear function of $\phi(X)=\left(X, X^2\right)$, that is, $e_{u, i}=\phi\left(X_{u, i}\right)^T \beta+\epsilon_{u, i}, \mathbb{E}\left[\epsilon_{u, i} \mid X_{u, i}\right]=0$. Then the IPS estimator with the balancing weights $w_{u, i}$ is unbiased**
> $$
> \sum_{(u, i): o_{u, i}=1} \hat w_{u, i} e_{u, i}=\sum_{(u, i): o_{u, i}=1} \hat w_{u, i} \phi(X_{u, i})^T \beta+\sum_{(u, i): o_{u, i}=1} \hat w_{u, i} \epsilon_{u, i} \approx \frac{1}{|\mathcal{D}|} \sum_{(u, i)\in\mathcal{D}} \phi(X_{u, i})^T \beta+0 \rightarrow \frac{1}{|\mathcal{D}|} \sum_{(u, i)\in\mathcal{D}} e_{u, i} \quad \text{(ideal loss)},
> $$
> where the approximately equal holds via the **balancing constraints** in the optimization problem.
>
> - **Instead, if $e_{u, i}$ is not a linear function of $\phi(X)=\left(X, X^2\right)$, then it is clear that the IPS estimator with balancing weights $w_{u, i}$ is not unbiased, and only balancing $\phi(X)=\left(X, X^2\right)$ is not enough.**
>
> ***
>
> Please kindly refer to the intuitive example we provided further above. We are glad to know that your concerns have been effectively addressed. We are very grateful for your constructive comments and questions, which helped improve the clarity and quality of our paper. Thanks again!

---

### Author Response · Authors · 2023-11-22
**General responses and manuscript revision summary**

Dear reviewers and AC,

We sincerely thank all reviewers and AC for their great effort and constructive comments on our manuscript. We know that we are now approaching the end of the author-reviewer discussion and apologize for our late rebuttal. During the rebuttal period, we have been focusing on these beneficial suggestions from the reviewers and doing our best to add several experiments and revise our manuscript. We believe our current carefully revised manuscript can address all the reviewers’ concerns.

As reviewers highlighted, we believe our paper tackles an important and relevant problem (**Reviewer 4dU2**, **Reviewer ptof**), introducing a novel interesting and perspective idea (**Reviewer CJCu**, **Reviewer ptof**). We also appreciate that the reviewers found the proposed methods reasonable (**Reviewer 4dU2**), intriguing and offers inspiration (**Reviewer CJCu**) with sound theoretical analysis (**Reviewer 4dU2**), as well as solid and convincing experiments (**Reviewer 4dU2**, **Reviewer ptof**).

Moreover, we thank the reviewers for pointing out the concerns regarding the motivation and clarity problems (**Reviewer CJCu**), as well as for the suggestions for investigating both types of kernel functions (**Reviewer CJCu**), the trade-off between effectiveness and efficiency (**Reviewer 4dU2**), and the scalability to the large-scale datasets and modern DL-based recommender systems (**Reviewer ptof**). In response to these comments, we have carefully revised and enhanced our manuscript with the following important changes with  the added experiments:

- [Reviewer CJCu] We **highlight our motivation** for carefully selecting the function class for balancing (in Introduction, Section 1).
- [Reviewer CJCu] We address the clarity issue by **discussing the rationality of the optimization problem** (in Sections 4.2 and 4.4), **adding definitions and examples of kernel functions** (in Section 4.3), **explaining how the proposed approach reduces the generalization bounds** (in Section 4.5), and **adding detailed proofs of theorems and corollaries** (in Appendix A).
- [Reviewer CJCu] We **add experiments using both Gaussian kernel and exponential kernel** to implement the proposed methods (in Table 1), as well as **clarify the missing kernel hyper-parameters tuning range** (in Section 5).
- [Reviewer 4dU2] We **add discussions of the relationships and distinctions between our work and prior research** (in the Related Work section).
- [Reviewer 4dU2] We **add experiments to analyze the trade-off between the effectiveness and the efficiency** of CBDR and ACBDR with detailed analysis (in Table 3, Appendix B).
- [Reviewer ptof] We **add experiments by taking Neural Collaborative Filtering (NCF) as the base model** on the **Product** dataset with detailed analysis (in Table 2, Appendix B).

These updates are temporarily highlighted in "$\textcolor{red}{red}$" for facilitating checking.

We hope our response and revision could address all the reviewers' concerns, and are more than eager to have further discussions with the reviewers in response to these revisions.

Thanks,

Submission9073 Authors.

---

### Meta-Review · Area_Chair_Rq8u · 2023-12-14

**Metareview:**

The paper proposed a a unified kernel-based method to balance functions on the reproducing kernel Hilbert space (RKHS) to mitigate selection bias in Recommender Systems. It provides theoretical analysis and guidance of balancing property under finite samples. Experimental results on three real-world datasets verify the proposed balancing methods.

Strength: reviewers find the idea of employing a linear combination of kernel functions for balancing to address selection bias interesting. The proposed method is supported by theoretical analyses and empirical validation.
Weakness: several reviewers asked for improved writing, e.g., missing definitions and proofs of theorems and corollaries. The authors were able to address them and convinced the reviewers to raise the rating through rebuttal.

**Justification For Why Not Higher Score:**

Several reviewers asked for improved writing. The authors were able to address them and convinced the reviewers to raise the rating through rebuttal.

**Justification For Why Not Lower Score:**

Reviewers find the idea of employing a linear combination of kernel functions for balancing to address selection bias inspiring. The proposed method is supported by theoretical analyses and empirical validation.

---

### Decision · Program_Chairs · 2024-01-16

Accept (spotlight)